# Electron Probe Microanalysis and Microscopy of Polishing-Exposed Solid-Phase Mineral Inclusions in Fuxian Kimberlite Diamonds

**Donggao Zhao**

Electron Microscope Laboratory of School of Dentistry, Department of Earth and Environmental Sciences, University of Missouri-Kansas City, Kansas City, MO 64108, USA; zhaodo@umkc.edu

**Abstract:** Solid-phase mineral inclusions in diamond (1–3 mm in diameter) from the No. 50 kimberlite diatreme of Liaoning Province, China, were exposed by polishing. A variety of silicate, carbonate and sulfide inclusions were recovered in the diamond. The common solid-phase inclusions are olivine, chromite, garnet and orthopyroxene; the rare phases include Ca carbonate, magnesite, dolomite, norsethite, pyrrhotite, pentlandite, troilite, a member of the linnaeite group, an unknown hydrous magnesium silicate and an Fe-rich phase. Abundance and composition of the solid-phase inclusions in diamond indicate that they belong to the peridotitic suite and are mainly harzburgitic. No eclogitic mineral inclusions were found in the diamond. The slightly lower Mg # of the olivine inclusions (peak at 93) than that of harzburgitic olivine inclusions worldwide (Mg # peak at 94), the higher Ni content (0.25–0.45 wt. %) of the olivine inclusions than those of olivine inclusions worldwide (0.30–0.40 wt. %), the higher Ti contents (up to 0.79 wt. %) in some chromite inclusions in diamond than those in chromite inclusions worldwide, the existence of carbonate inclusions in diamond, and the possible presence of hydrous silicate phases in diamond all indicate a metasomatic enrichment event in the source region of diamond beneath the North China craton, suggesting that the diamond probably formed by solid-state growth under metasomatic conditions with the presence of a fluid. Solid-state growth of diamond is also supported by abundant graphite inclusions in the diamond. Sulfide inclusions in diamond often coexist with chromite and olivine or are rich in Ni content, indicating that the sulfide inclusions belong to the peridotitic suite. From the chemical compositions, most sulfide inclusions in diamond from the No. 50 kimberlite were probably trapped as monosulfide crystals, although some may have been entrapped as melts.

**Keywords:** electron probe microanalysis and microscopy; solid-phase mineral inclusion; Fuxian kimberlite; metasomatic event; diamond formation

## 1. Introduction

Natural diamond often contains mineral or fluid inclusions that provide information about the chemical and physical environment in the upper mantle. Primary mineral inclusions in diamond can be assigned to either peridotitic (P-type) or eclogitic (E-type) paragenesis [1,2], corresponding to the two major mantle xenolith types; they may also be divided into two categories: primary and secondary. Primary inclusions formed before or during growth of the diamond hosts. They usually do not have fractures connected to the outside of the host and are completely contained in diamond. Primary inclusions have been protected by the diamond host since they were entrapped. On the other hand, secondary inclusions, those that formed after the formation of the diamond hosts or by modification of primary inclusions exposed to the surrounding environment through fractures in the diamond crystal, usually have a fracture connected to the outside of diamond host. Secondary inclusions provide ambiguous records of multi-stage processes, for example,

during eruption of the host kimberlitic magma. Knowledge of chemical composition of inclusions allows reconstruction of diamond growth conditions (e.g., P-T-$fO_2$) and thus contributes to an understanding of upper mantle processes. Primary inclusions usually occur as crystals, albeit with negative crystal faces, or as amorphous substances. In terms of mineral species, inclusions in diamond include silicate, sulfide, and more rarely carbonate, oxide, carbide, and native elements (graphite, diamond, metals and alloys). A favorable tectonic environment for diamond is present in three cratons in China, the North China or Sino-Korean, the Yangtze, and the Tarim Cratons. In the North China craton, hundreds of kimberlite diatremes and dikes have been located [3]. The kimberlite cluster in Fuxian (now called Wafangdian), Liaoning Province is at the south end of the East Liaoning Uplift of the North China craton and has more than 100 kimberlite bodies. Previous studies on mineral inclusions in diamond from the Fuxian kimberlite mostly focused on specific and unusual types of inclusions [3–21] , for example, silicon carbide in diamond by [5], high-Cu and high-Cl inclusions by Chen et al. [13], a carbon-rich multiphase inclusion by Wang et al.[12], syngenetic inclusion geochemistry by Harris et al. [16] and Meyer et al. [17], and native metals and alloys by Gorshkov et al. [20]. Some of these studies are qualitative, without compositional data, and some do not indicate whether the inclusions are primary or secondary. To systematically study mineral inclusions in diamond from the No. 50 kimberlite diatreme in the North China craton, we used the polishing technique to expose mineral inclusions in diamond. A total of 355 diamond crystals with inclusions were examined. Among them, more than 100 diamonds were polished to expose the mineral inclusions for further studies. These diamonds were selected from thousands of diamonds produced by the Liaoning Sixth Geological Exploration Team.

## 2. Geology of the Fuxian Kimberlite

The Fuxian kimberlites are located about 60 km west of Kaiyuan-Yingkou fault and 30–40 km east of the Tanlu Fault, a major NNE-trending transcurrent structure in East China extending thousands of kilometers from Northeastern to Central China. In the Fuxian area, from north to south, there are three parallel kimberlite zones. The northern zone is medially rich in diamond and consists of numerous kimberlite pipes and dikes, including China's largest kimberlite body, the No. 42 diatreme (4000 m² on the surface). The central zone, also called Toudaogou zone, is most rich in diamond. The famous No. 50 diatreme is situated in the central zone. In the neighboring gullies, some dikes and diamond placers are present. The other kimberlite bodies in the central zone include No. 51, 68 and 74 diatremes and some dikes. The southern zone is poor in diamond and consists of small kimberlite bodies. Country rocks of the Fuxian kimberlite are the Proterozoic Nanfen and Qiaotou Formations. The Nanfen Formation consist mainly of shale with some siltstone and occurs in the southwest part of the kimberlite area. The Qiaotou Formation consists predominantly of thick quartz sandstone with some thin siltstone and occurs in the whole area. The highest stratigraphic level of sediment intruded by the Fuxian kimberlite is the Maozhuang Formation of middle Cambrian age. It was estimated that about 1000–1500 m of metasediments were eroded away at the No. 50 diatreme, while at the No. 42 diatreme, the depth of metasediments eroded could be more than 1500 m [3] [3]. The Fuxian kimberlite is believed to have been emplaced about 400 to 500 Ma ago [3]. Diabase dikes or sills also occur in this area and sometimes cut through the kimberlite bodies (Liaoning Sixth Geological Team 1990, unpublished manuscript).

The No. 50 kimberlite diatreme occurs as an irregular rhombus on the ground. The long axis of the diatreme is in an east–west direction and extends for about 275 m; the short axis is in south–north direction and extends for about 65 m. The diatreme dips 85° to the southeast. The upper and middle parts of the No. 50 diatreme consist mainly of kimberlite tuff breccia. Kimberlite from the No. 50 and No. 42 diatremes has inequigranular texture. Alteration such as serpentinization and carbonatization of the kimberlite is extensive and strong. Primary phenocryst/xenocryst minerals include olivine, phlogopite, pyrope, chromite, diamond, and zircon. The matrix consists primarily of late-stage

olivine, phlogopite, carbonate, apatite, magnetite, chromite, anatase, perovskite, rutile, zircon, wollastonite and diopside [3].

### 3. Sample Preparation

Typically, most of mineral inclusions in the diamond are extracted by crushing or burning diamond hosts. The shortcomings of these two techniques include contamination and oxidation of mineral inclusions and difficulties in recovering small inclusions. Recently, polishing has been employed to reveal mineral inclusions in diamond [22]. The size of the inclusions in diamond exposed by polishing may be as small as 5 μm, whereas inclusions extracted from diamond by mechanical crushing are generally larger than 30 μm in diameter [23,24] (.

The sample preparation procedure is as follows. An inclusion-bearing diamond is first cleaned in ethanol or alcohol ultrasonically three times (use acetone to clean the crystal bond). The shape, size, and color of a diamond and its inclusions (if any) are examined and described with an optical microscope. A clean specimen mold or bottle cap is used to produce an epoxy resin disk with a diamond crystal. The mold is then labeled with the sample number and some mold release is applied to help remove resin from the mold after its solidification. The cleaned diamond is then oriented on a piece of double-sided sticky tape on the bottom of the mold so that the (111) face of a diamond is not parallel to the bottom of the mold. This orientation will avoid polishing of the hardest (111) face of the diamond. A second way to avoid polishing of the (111) face is to polish the sample at a direction that is not parallel to the bottom of the mold. Liquid resin (epoxy : hardener = 5:1) is gently poured into the mold and vacuumed to get rid of bubbles. After ~24 h, the solidified resin disc with diamond can be removed from the mold. The resin disc is then polished carefully by using diamond-impregnated polishing wheels (diamond size 70, 30 or 15 μm). A 15-μm diamond wheel is effective enough for final polishing for chemical characterization using electron probe microanalysis (EPMA), also called electron microprobe analysis (EMPA), although a 6-μm diamond wheel is better. A brand new diamond wheel polishes diamond crystal very quickly. The detailed sample preparation procedure was documented by Zhao [25].

### 4. Analytical Methods

A variety of analytical techniques, including backscattered electron (BSE) imaging from scanning electron microscopy (SEM), qualitative analysis and mapping of X-ray energy dispersive spectroscopy (EDS), quantitative analysis from EPMA wavelength dispersive spectroscopy (WDS), and micro-Raman spectroscopy, were employed to characterize mineral inclusions and diamond hosts. Species of mineral inclusions can be qualitatively identified using EDS because many phases have a characteristic or unique EDS spectrum. Such identification is routinely performed with SEM and EPMA and the results are usually reliable, especially when combined with other methods, such as optical microscopy and EPMA WDS quantitative analysis. BSE and EDS data were acquired on the Hitachi S-570 SEM or the Cameca CAMEBAX electron microprobe at the University of Michigan. Mineral inclusions, mostly olivine, to be identified by micro-Raman were not polished to the surface since micro-Raman beam has ability to penetrate into diamond host. The micro-Raman spectra were acquired using a Renishaw Raman microscope at the University of Michigan.

Compositions of mineral inclusions were determined quantitatively on the Cameca CAMEBAX electron microprobe equipped with four spectrometers. Element concentrations were determined by WDS with a PAP correction routine. Most mineral inclusions were analyzed with a focused beam in a spot mode, but carbonates and water-bearing phases were analyzed with a raster mode. Peak and background counting times were set at 30 and 15 s, except for Si in chromite, which was counted for 120 s to increase precision. Background positions were adjusted when two backgrounds were significantly different. To examine homogeneity, multiple points were acquired for each phase.

Natural and synthetic materials were used as standards. For analysis of silicate minerals (such as olivine, pyroxene, and garnet), an accelerating voltage of 15 kV and a beam current of 10 nA were used. The standards used are clinopyroxene (from Irving) as a standard for Si and Ca, olivine for Mg, ferrosilite for Fe, almandine (from Ingamells) for Al, uvarovite for Cr, geikielite for Ti, rhodonite for Mn, NiS for Ni, and jadeite (from ANU) for Na. For analysis of spinel/chromite analyses, ferrosilite (synthetic) was used as standard for Si, geikielite (synthetic) for Mg and Ti, chromite for Al, $V_2O_5$ (synthetic) for V, $Cr_2O_3$ (synthetic) for Cr, rhodonite (Broken Hill) for Mn, hematite (Elba) for Fe, and NiS (synthetic) for Ni. Spinel was analyzed at an accelerating voltage of 10 kV and a beam current of 45 nA. The lower voltage was used to reduce the potential secondary fluorescence effect of Cr by Fe [26]. Chromite inclusions in diamond may contain higher Si content than other chromites. To increase the precision and accuracy of Si measurements, the counting time for Si in chromite inclusions was intentionally set at 120 s. For sulfide, FeS (synthetic) was used as standard for S and Fe, Scott chalcopyrite for Cu, NiS (synthetic) for Ni, ZnS (synthetic) for Zn, MnS (synthetic) for Mn and CoS (synthetic) for Co. Sulfide was analyzed at an accelerating voltage of 20 kV and a beam current of 10 nA or at 10 kV and 20 nA if K and Cr were included. The lower voltage was also used to reduce the fluorescence effect of Fe by Ni.

### 5. Features of Diamond Hosts and Their Mineral Inclusions

The diamond crystals were optically examined for color, size, crystal form, fractures, and mineral inclusions. Most of the diamonds from the No. 50 kimberlite diatreme are colorless and transparent; some diamond crystals are light yellow, light brownish yellow, light gray, milky white or brown; occasionally, diamond crystals are light blue, light yellowish green, pink or black [3]. The size of the diamond ranges from 1 to 3 mm in diameter. The diamond is mainly single crystal. The common forms are step-like octahedron, curved-face rhombic dodecahedron, perfect octahedron, and combined forms; twinned diamond (for example, macles) or aggregated crystals were also identified but are rare. The partial resorption of octahedra along edges can be observed. Some single crystals are distorted while others possess perfect crystal forms. Simple and combined crystal forms of diamonds were recognized under the binocular microscope and by using a goniometer. The common crystal forms include the following: octahedra, o{111}, usually layered; dodecahedra, d{110}, curved or layered faces. Other shapes, such as cube, hexoctahedron, tetrahedron, etc. are rare [15].

About 40% of the diamond crystals contain graphite, and approximately 0.8% diamond crystals have silicates and other inclusions [3]. A preliminary visual inspection under a binocular microscope showed that the most common mineral in the diamond is graphite in the form of gray or black flakes. Inclusion phases were visually examined for associated fractures to the diamond surface. Most mineral inclusions in diamond were first exposed on the polished surface of diamond using the polishing procedure described above. After optical examination, mineral inclusions were characterized by SEM, micro-Raman, and EPMA, if the inclusions were large enough. Table 1 shows the inclusions identified in the diamonds of the No. 50 kimberlite. Except for graphite, these mineral inclusions are almost exclusively peridotitic, with olivine being the most abundant inclusion, followed by chromite, pyrope, sulfide, orthopyroxene, Ca-carbonate, magnesite, and a Fe-dominant phase (Figure 1). There are often two or more, same or different inclusions in one diamond crystal. Inclusion shapes are generally octahedral, often flattened or elongated (e.g., Figure 1a), showing that the negative crystals of the inclusion morphology had been imposed by the diamond host [27].

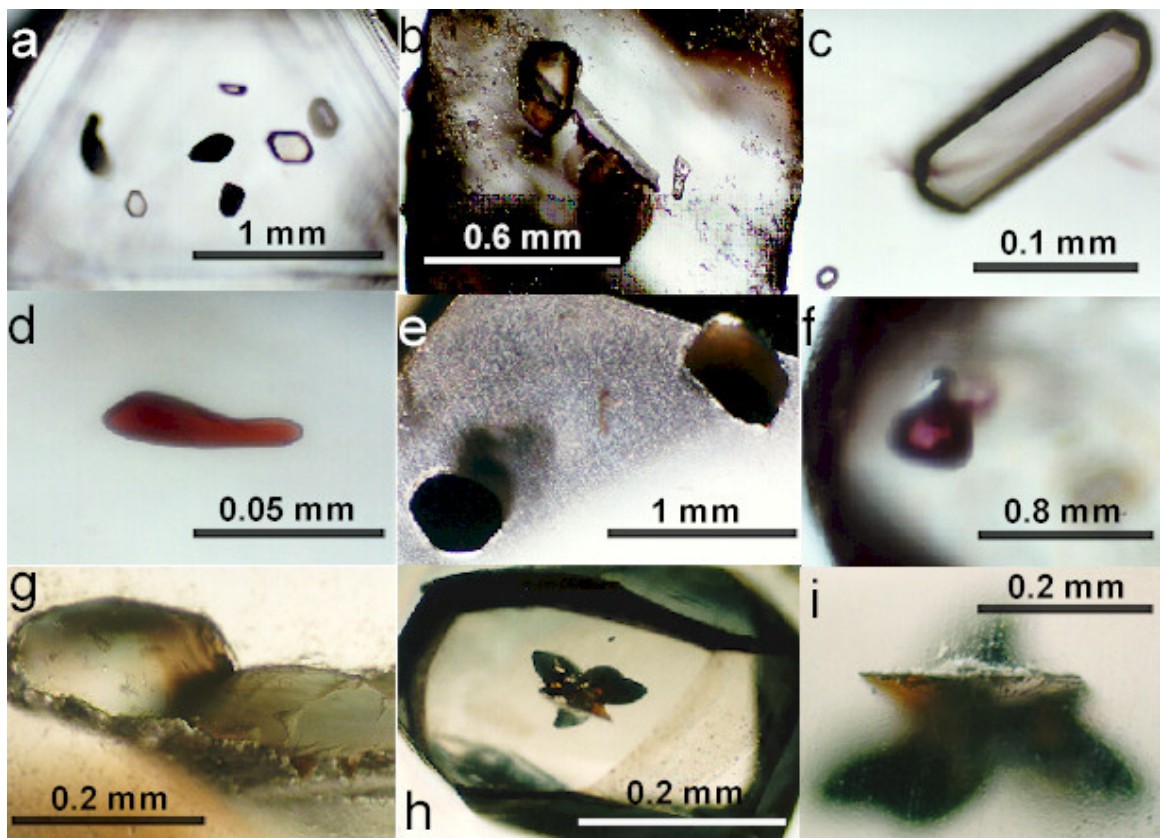

**Figure 1.** Transmitted plane-polarized light images of diamond hosts and their mineral inclusions. (**a**) Multiple inclusions in the diamond LN50D12 with at least 7 inclusions (4 olivine and 3 chromite). (**b**) Inclusions of two larger and one smaller olivine in the diamond LN50D02. There are some black platelets over the surface of the small euhedral elongated olivine inclusion. (**c**) Two olivine inclusions distributed along a NE direction in the diamond LN50D04. (**d**) An irregular chromite inclusion in the diamond LN50D04. (**e**) Two chromite inclusions in the diamond LN50D07. The upper right chromite inclusion is only partly included in the diamond host. (**f**) Garnet (purple, on the left) and olivine (lower right corner) inclusions in the diamond LN50D10. (**g**) Two orthopyroxene inclusions in the diamond LN50D40. (**h**) Butterfly-like graphite inclusion in the diamond LN50D38. (**i**) Enlarged image of the graphite inclusion in (**h**). X-ray mapping of the fracture shows no Si, indicating that the inclusions are not SiC.

**Table 1.** Mineral inclusions identified in diamonds from the No. 50 kimberlite diatreme.

| Protogenetic/syngenetic | | Epigenetic | Uncertain |
|---|---|---|---|
| **Ultramafic** | **Eclogitic** | | |
| forsterite[*] | Omphacite [3,20] | calcite[*] | phlogopite [3,20] |
| enstatite[*] | coesite (quartz) [9] | magnesite[*] | magnetite [20] |
| Diopside [3,15,20] | rutile [20] | graphite[*] | apatite [3,20] |
| pyrope[*] | ilmenite [20] | | moissanite [5] |
| spinel[*] | graphite[*] | | graphite[*] |
| ilmenite [20] | | | native iron [9,20] |
| sulfides[*] | | | Fe-dominant phase[*] |
| zircon [20] | | | Cr-Fe-Ni alloy [20] |
| diamond[*] | | | |
| graphite[*] | | | |
| native iron [9,20]? | | | |
| native chromium [20]? | | | |

magnesite[*]

Ca-carbonate[*]

hydrous Mg-silicate[*]

[*]This work.

Olivine inclusions are very common (Figures 1b,c and 2). Multiple olivine inclusions can occur in the same diamond or associated with other minerals. For example, in the diamond LN50D04, there are two olivine inclusions and one chromite inclusion (Figure 2c). Sometimes two olivine inclusions have different orientations in a diamond. For example, in the diamond LN50D03 (Figure 2b), the c axis of the larger olivine is perpendicular to the polished section, while the c axis of the smaller one is close to parallel to the polished section. The olivine inclusions are euhedral (Figure 2b,d), elongated (Figure 2g,i), subhedral (Figure 2h), or anhedral (Figure 2f). The form of an olivine inclusion is often strongly constrained by the crystal form of the diamond host. The bent feature of the olivine crystal in the diamond LN50D45 (Figure 2g) might be evidence of such constraint. The ring structure surrounding the olivine in the diamond LN50D04 (Figure 2d) may be the result of strain between the inclusion and the diamond host.

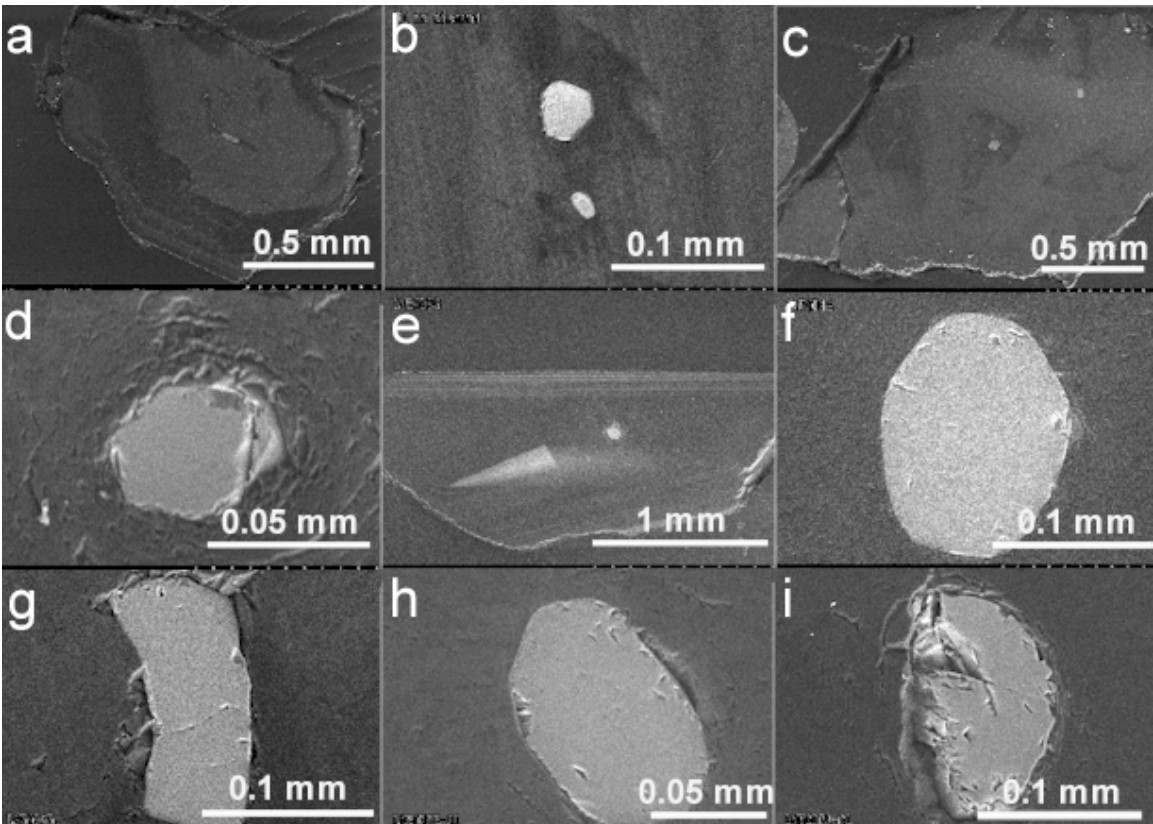

**Figure 2.** Backscattered electron images of diamond hosts and their olivine mineral inclusions (reproduced from [21] with permission from Cambridge University Press). (**a**) Diamond LN50D03 and an elongated olivine inclusion. Note the cathodoluminescent pattern of the diamond host. (**b**) A second polished surface of the diamond LN50D03 with two euhedral olivine inclusions. The c axis of the large olivine is perpendicular to the polished section, while the c axis of the small olivine is close to parallel to the polished section. (**c**) Two olivine inclusions and one chromite inclusion at the upper right corner in the diamond LN50D04. Note the cathodoluminescent pattern surrounding the olivine at the center. (**d**) The euhedral olivine inclusion at the center of the diamond LN50D04. Note the ring structure surrounding the olivine inclusion, a possible result of strain between the inclusion and the host. (**e**) Diamond LN50D39 with an olivine inclusion and a triangular cathodoluminescent pattern. (**f**) One anhedral olivine inclusion in the diamond

LN50D44. (**g**) One euhedral olivine inclusion in the diamond LN50D45. Note the bent feature of the inclusion that was possibly constrained by the crystal form of diamond host or simply trapped as such during the formation of the diamond. (**h**) One subhedral olivine inclusion in the diamond LN50D55. (**i**) One elongated olivine inclusion in the diamond LN50D68.

Chromite inclusions are also common (Figures 1d,e and 3). Like olivine, there are often two or more chromite inclusions in a diamond (LN50D07, Figure 3a). The crystals of chromite inclusions are subhedral (LN50D12, Figure 3d) or euhedral (LN50D58, Figure 3f). Interestingly, small chromite inclusions tend to form perfect crystal forms, which may indicate that the chromite crystals form negative crystal faces imposed by the diamond host, and are located close to the center of the host (Figure 3e), while larger chromites are usually anhedral or subhedral and are distributed near the edge of the diamond (Figure 3a). In the diamond LN50D07, there are three chromite inclusions, of which two touch each other and the third one is separate (Figure 3a). The smallest chromite occurs in the outermost zone of cathodoluminescence, but all the three chromites are close to the edge of the diamond, indicating that the inclusions were trapped at the later stages of diamond growth. The euhedral chromite at the center of cathodoluminescent pattern of the diamond LN50D58 (Figure 3e) indicates that the inclusion was trapped at the early stage of diamond growth.

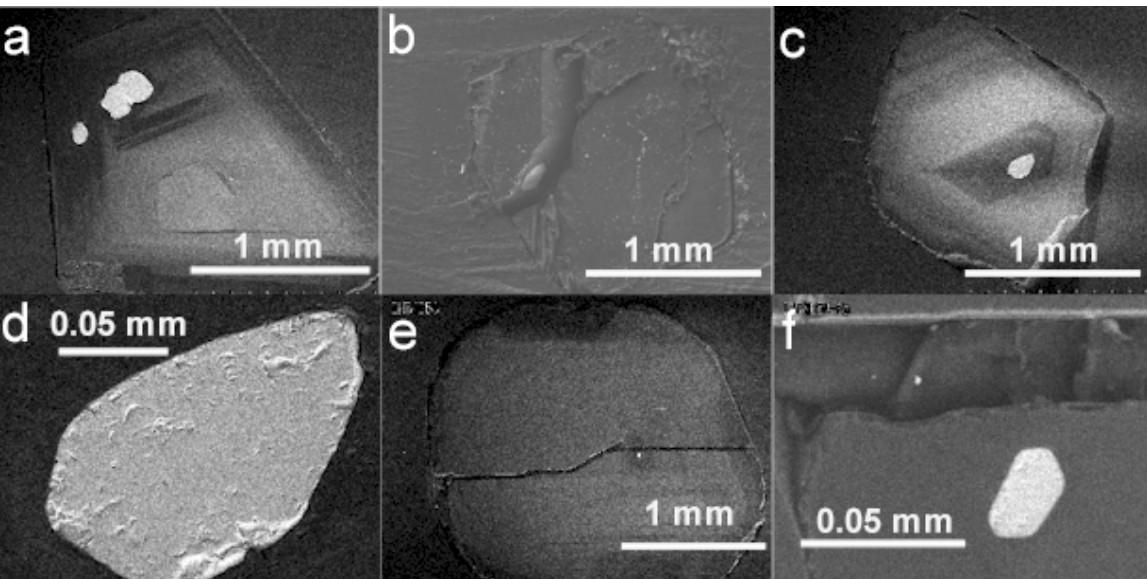

**Figure 3.** BSE images of diamond hosts and their chromite mineral inclusions. (**a**) Diamond LN50D07 and three chromite inclusions distributed along the edge of cathodoluminescent pattern. (**b**) Diamond LN50D45 and a chromite inclusion in the broken fracture developed during polishing. (**c**) Diamond LN50D12 and a chromite inclusion at the center of cathodoluminescent pattern. (**d**) The chromite inclusion in (**c**) at higher magnification. (**e**) Diamond LN50D58 and a tiny (30 μm) chromite inclusion. The diamond host was broken during polishing. (**f**) The euhedral chromite inclusion in (**e**) at higher magnification.

Garnet was identified in a few diamond crystals (Figures 1f and 4). The diamond LN50D10 has two garnet inclusions, one of which is subhedral (Figure 4b). A garnet inclusion in the diamond LN50D13 (Figure 4c) was broken, and there are fractures connected to the outside, but its composition is similar to those of other garnet inclusions, indicating that it did not form or re-equilibrate after the formation of the diamond host.

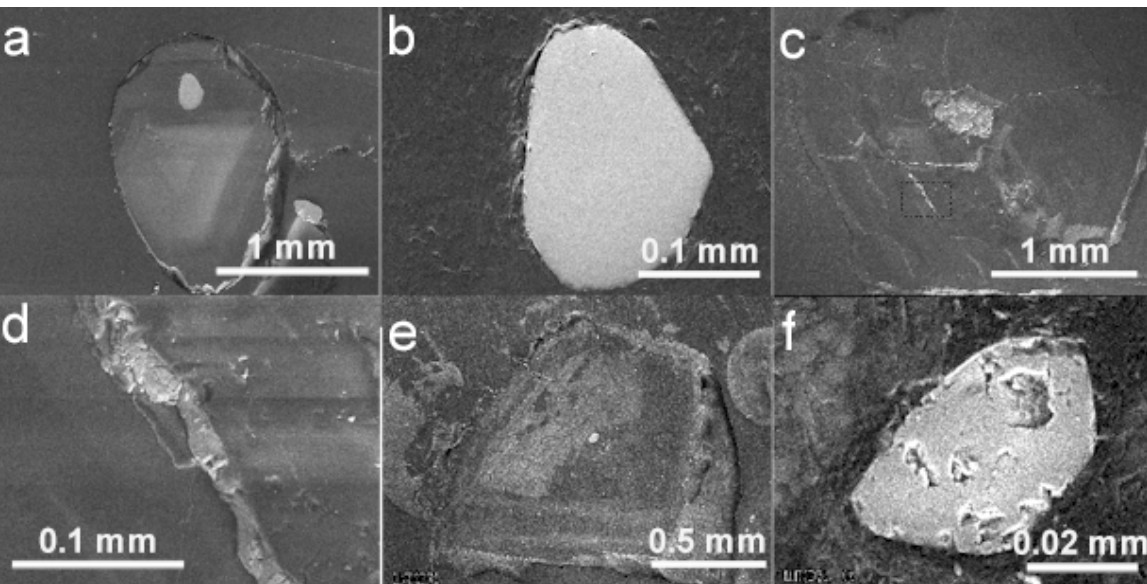

**Figure 4.** BSE images of diamond hosts and their garnet mineral inclusions. (**a**) Diamond LN50D10 and a garnet inclusion. (**b**) The garnet inclusion in (**a**) at higher magnification. (**c**) Diamond LN50D13 and a garnet inclusion with fractures connected to the outside. Composition of the relatively large inclusion suggests that it was not formed after the formation of diamond host. (**d**) Carbonate inclusions in a fracture in the diamond LN50D13 at higher magnification. (**e**) Diamond LN50D71 and a garnet inclusion. Note the complex cathodoluminescent pattern. (**f**) The garnet inclusion in (**e**) at higher magnification.

Four orthopyroxene inclusions were identified. There are two touching orthopyroxene crystals in the diamond LN50D40 and each is approximately 200 μm (Figures 1g and 5a,b). One green orthopyroxene that is 350 μm long was found in the diamond LN50D65.

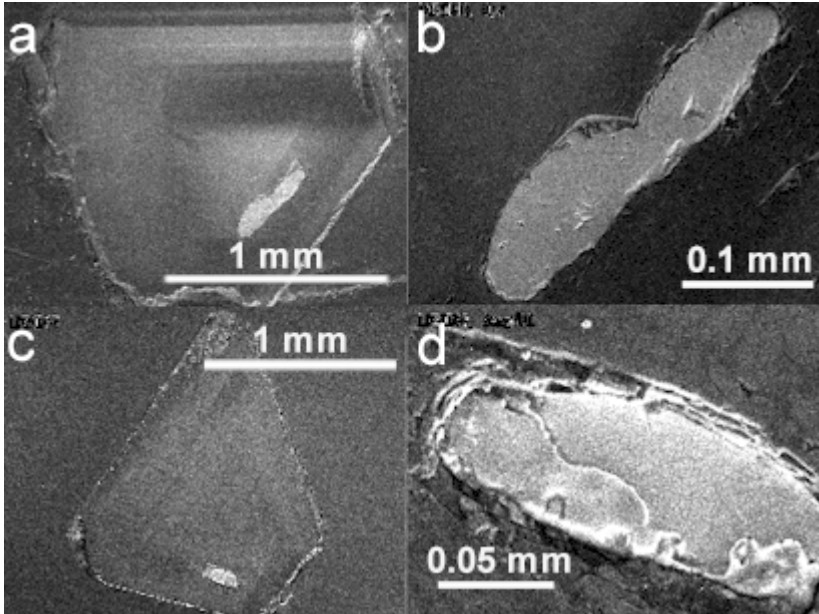

**Figure 5.** BSE images of diamond hosts and pyroxene and other silicate mineral inclusions. (**a**) Diamond LN50D40 and two coexisting orthopyroxene inclusions, also showing cathodoluminescent pattern. (**b**) The orthopyroxene inclusions in (**a**) at higher magnification. (**c**) Diamond LN50D67 and an unknown silicate inclusion. (**d**) The unknown silicate inclusion in (**c**) at higher magnification.

Two primary Ca carbonate inclusions were identified in the LN50D11 and LN50D97, respectively, by SEM. One of them was approximately 80 μm in size and was destroyed

during further polishing for EPMA (diamond LN50D11, Figure 6a,b). An irregular magnesite inclusion was identified in the diamond LN50D29. Secondary Ca carbonate and magnesite were identified within a fracture in the diamond LN50D13 (Figure 4d).

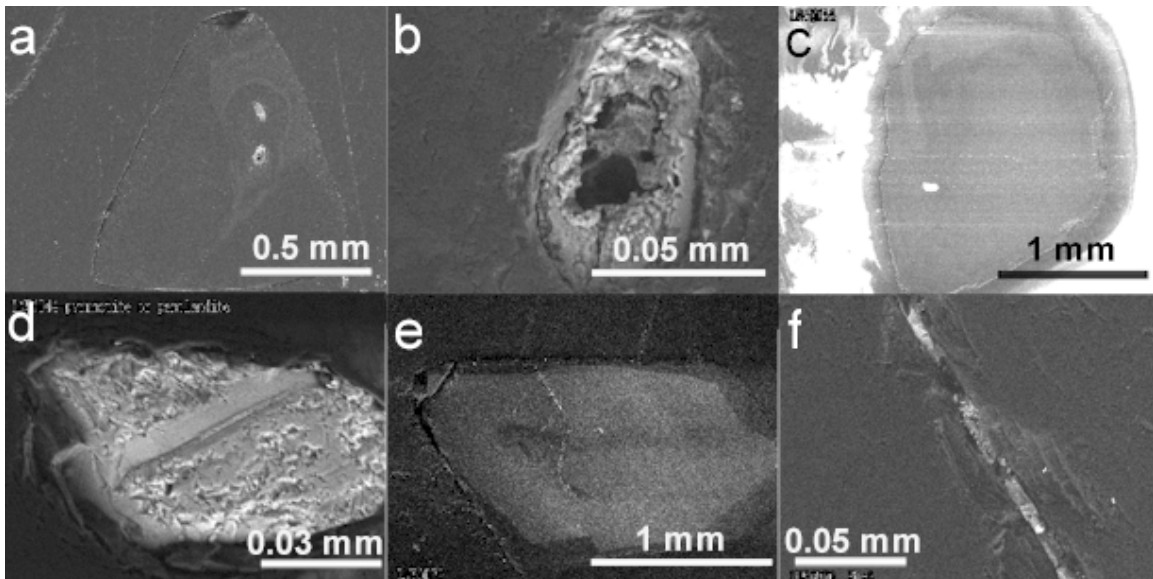

**Figure 6.** BSE images of diamond hosts and carbonate and sulfide mineral inclusions. (**a**) Diamond LN50D11 with one Ca-carbonate and one olivine inclusion. (**b**) The Ca-carbonate inclusion in (**a**), partly destroyed, at higher magnification. (**c**) Diamond LN50D42 and a sulfide inclusion. (**d**) The sulfide inclusion in (**c**) at higher magnification. (**e**) Diamond LN50D70 and a protogenetic or syngenetic olivine inclusion close to the center. A fracture extended to the surface of diamond cuts through the cathodoluminescent pattern, indicating that it was developed after the formation of the diamond host. In the top part of the fracture, an epigenetic pyrite inclusion was identified. (**f**) The epigenetic sulfide inclusion in (**e**) at higher magnification.

Sulfide inclusions were recovered in diamond samples LN50D04, LN50D32, LN50D37, LN50D42, and LN50D70 (Figure 6c,f). Most sulfide inclusions in diamond are primary (Figure 6c,d) and range from 20 to 50 µm in size. Primary sulfide inclusions are usually associated with internal fractures in diamond, and the fractures are filled with graphite. One secondary sulfide inclusion was found in a fracture that extended to the diamond surface (Figure 6e,f). This secondary sulfide inclusion is about 15 µm long, smaller than most primary sulfide inclusions.

Graphite inclusions are the most abundant, and occur as flakes or as aggregates of flakes, like a butterfly, in internal fractures or fractures extended to the outside (Figure 1h). On the crystal faces of some olivine inclusions, there are tiny (1–2 µm) black platelets (Figure 1b). Black dendritic materials are also found in the interfaces of diamond and inclusion. According to Harris [28], these black platelets on the surface of mineral inclusions are graphite. A special effort was made to find moissanite (SiC), an inclusion reported in the No. 50 diatreme diamond [5], but none was found. X-ray mapping of the fracture filled with graphite and sulfide in the diamond LN50D32 shows no Si counts, indicating that there is not any SiC inclusion in this fracture.

Other inclusions extracted by polishing include an unknown (hydrous) magnesium silicate (LN50D67, Figure 5c, 5d), an Fe-rich phase (LN50D09), and diamond (LN50D35 and LN50D36). A diamond is included in another diamond, indicating that diamond is of multistage genesis. Diopside, omphacite, phlogopite, rutile, zircon, magnetite [20], ilmenite, apatite [3,20], coesite [9], silicon carbide [5], native iron [9,20], native chromium and Cr-Fe-Ni alloy [20] have also been reported as inclusions in the Fuxian diamonds, but they were not encountered in this work (Table 1).

## 6. Results and Discussion

### 6.1. BSE Images and Cathodoluminescence of Diamond Hosts

The BSE images of polished sections of diamond often show certain cathodoluminescent patterns (e.g., Figure 2a,c,e). The internal structures of diamond revealed by cathodoluminescence include the following: (1) zoned patterns, e.g., sample LN50D3 (Figure 2a) has two distinctive bright and dark zones on the BSE image; (2) triangular patterns, e.g., in samples LN50D39 (Figure 2e) and LN50D40 (Figure 5a). The cathodoluminescent patterns surrounding the inclusions at the centers of diamonds LN50D04 (Figure 2c), LN50D12 (Figure 3c) and LN50D58 (Figure 3e) suggest that some inclusions may have served as the seed of diamond growth. The internal structure of diamond revealed by cathodoluminescence and the relative position of inclusions in the diamond host allow the variation of the chemistry of the inclusion over time during the growth of the host diamond to be investigated. Mineral inclusions in the same cathodoluminescent zone of a diamond may have formed at the same time and under the same P-T conditions. Therefore, meaningful P-T conditions calculated from mineral inclusion assemblages in the diamond host must come from the same cathodoluminescent zone.

### 6.2. EDS and X-ray Mapping of Mineral Inclusions

EDS qualitative identification of mineral inclusions could be affected by sample preparation and its geometric relation with the EDS detector. A phase to be identified must be on the surface and there must be no substance blocking the characteristic X-rays reaching the detector. If an inclusion is located in an unevenly broken fracture or is blocked by the diamond host, characteristic X-rays from an inclusion may not be able to reach the detector, due to the geometry between detector and sample. For example, the diamond host LN50D45 was broken during polishing and an inclusion was exposed on the downward fracture surface (Figure 3b). To avoid loss of the inclusion, the sample was subsequently imaged and identified with SEM and EDS without further polishing. Initially, only Cr and C peaks were identified in the EDS spectra of the inclusion, seemly suggesting that the inclusion phase is native Cr or CrC. However, when the same sample was coated with carbon and put into the SEM chamber again with 180° rotation horizontally, elements O, Mg, Al, Cr and Fe were detected in the EDS in the upper part of the inclusion, indicating the inclusion is chromite. However, in the lower part of the inclusion, still only Cr and C peaks were detected. X-ray mapping of elements in the chromite inclusion also did not detect light elements in the lower part of the inclusion. Clearly an EDS spectrum can be strongly affected by sample geometry and position of the detector relative to the sample. The reported native Cr in carbonado and diamond by [20,29] contains Cr peak and tiny O and Al peaks, suggesting that chromite might have been misidentified as Cr metal.

X-ray mapping is often used to determine homogeneity or heterogeneity of a phase or intergrowth. In Figure 7, X-ray maps of sulfide inclusions in diamond show a heterogeneous distributions in S, Fe, Ni and Cu, consistent with the EPMA results.

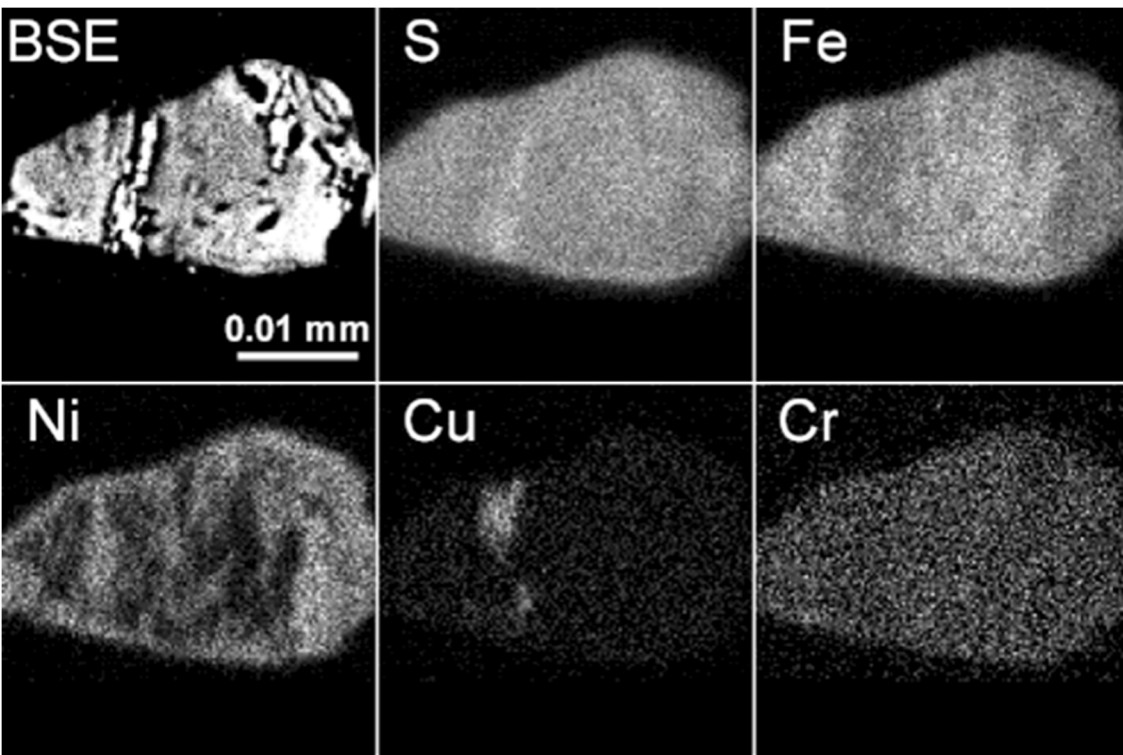

**Figure 7.** X-ray mapping of elements in a sulfide inclusion in the diamond LN50D04. Fe, Ni, Cu and S are heterogeneously distributed and Cu is much enriched in some areas.

*6.3. Micro-Raman Spectra of Mineral Inclusions*

Micro-Raman spectroscopy was used to identify mineral species of inclusion phases in diamond qualitatively. The micro-Raman technique provides structural information of phase and allows distinction of different polymorphs of the same composition, for example, olivine, wadsleyite and ringwoodite with $(Mg,Fe)_2SiO_4$. It also has the capability to penetrate into the diamond host. Therefore, an inclusion that is still completely included in diamond but close to the polished surface may be identified by its micro-Raman spectrum. Three micro-Raman spectra were obtained for three olivine inclusions (Figure 8): one olivine inclusion is on the polished surface of the diamond (LN50D68); a second olivine inclusion is below the polished surface (LN50D73); a third olivine is also below the polished surface (LN50D96). These inclusions (Figure 8) have clearly visible peaks around 820 and 850 cm$^{-1}$, consistent with the main olivine Raman peaks [30](, indicating that the $(Mg,Fe)_2SiO_4$ phases identified by EPMA (see below) are not wadsleyite or ringwoodite. Wadsleyite and ringwoodite have main peaks at 721 and 918 cm$^{-1}$, and 796 and 841 cm$^{-1}$, in the range of 500 to 1300 cm$^{-1}$, respectively [31]. In addition, the olivine inclusion exposed on the surface (Figure 8a) has stronger peaks around 820 and 850 cm$^{-1}$ than those below the polished surface (Figure 8b,c). Some low EPMA totals of olivine inclusions are probably caused by the lower surface of inclusion relative to the diamond host due to polishing.

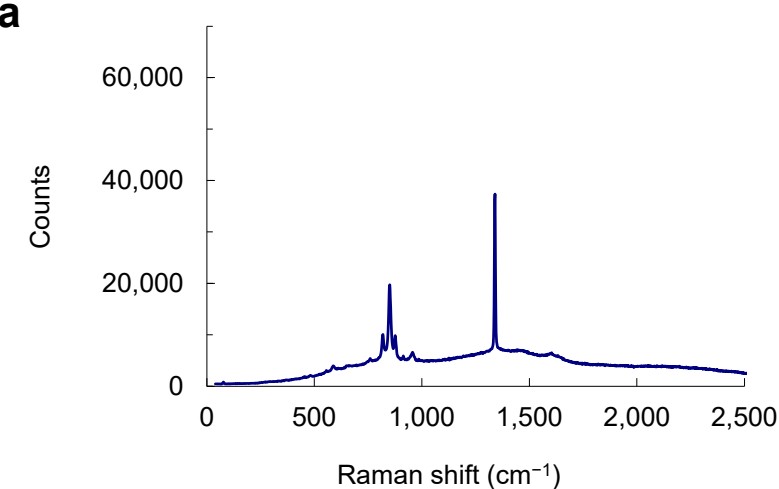

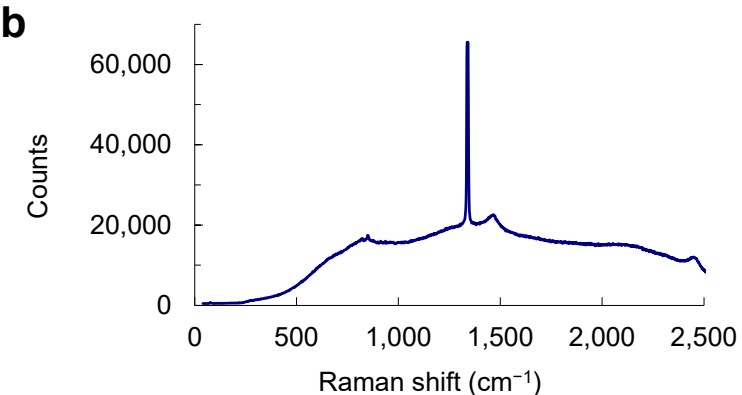

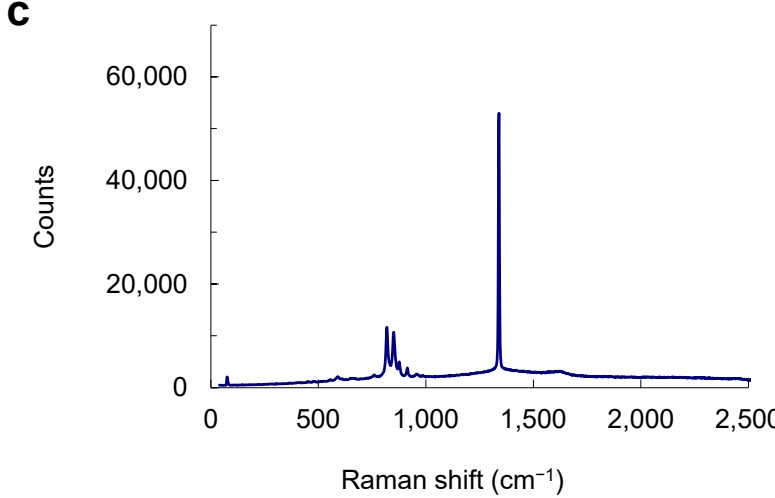

**Figure 8.** Polarized, single-crystal Raman spectra of olivine inclusions. (**a**) On the polished surface of the diamond LN50D68. (**b**) Below the polished surface (LN50D73). (**c**) Below the polished surface with a low EPMA total (LN50D96). Olivine inclusions with normal and low EPMA totals show the same or similar micro-Raman spectra.

### 6.4. Chemical Compositions of Peridotitic Mineral Inclusions

Chemical compositions of mineral inclusions are presented in Tables 2 to 8. All mineral inclusions are homogeneous except for sulfide ones.

**Olivine.** Approximately 47 olivine inclusions were analyzed in this work (Table 2) and additional EPMA analyses of olivine inclusions from the Fuxian kimberlites are also available in the literature [3,16,17]. Olivine inclusions typically have a high Mg # of 92–94. Mg # is defined as $100Mg/(Mg + Fe^{2+})$ by atom where all Fe is assumed to be $Fe^{2+}$ for olivine and $Fe^{3+}$ is calculated in the case of chromite. The peak position for the Mg # of olivine is 93, a value that is slightly lower than average Mg # of 94 for worldwide olivine inclusions [1,32,33], but similar to those of olivine inclusions from the Akwatia diamonds [24]. Only two olivine inclusions have a Mg # that is outside the range of 92–94 (Figure 9a). The NiO contents of the olivine inclusions are usually in the range of 0.25 to 0.45 wt. % and are concentrated around 0.4 wt. % (Table 2; Figure 9b). One exceptional value is for the olivine from the diamond LN50D62, which has up to 0.80 wt. % NiO. Due to the wide NiO range (0.25–0.45 wt. %), the constant Ni content assumption for mantle olivine for the Ni-in-garnet thermometer may not always be valid [34,35].

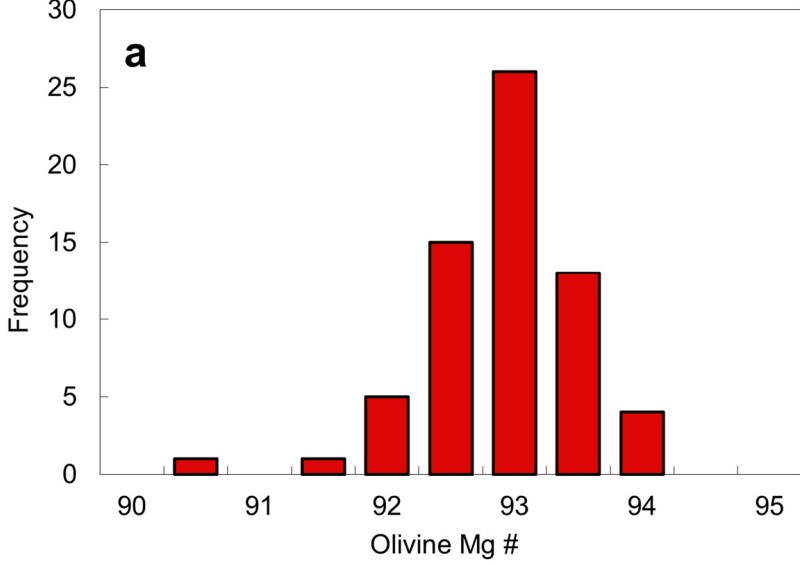

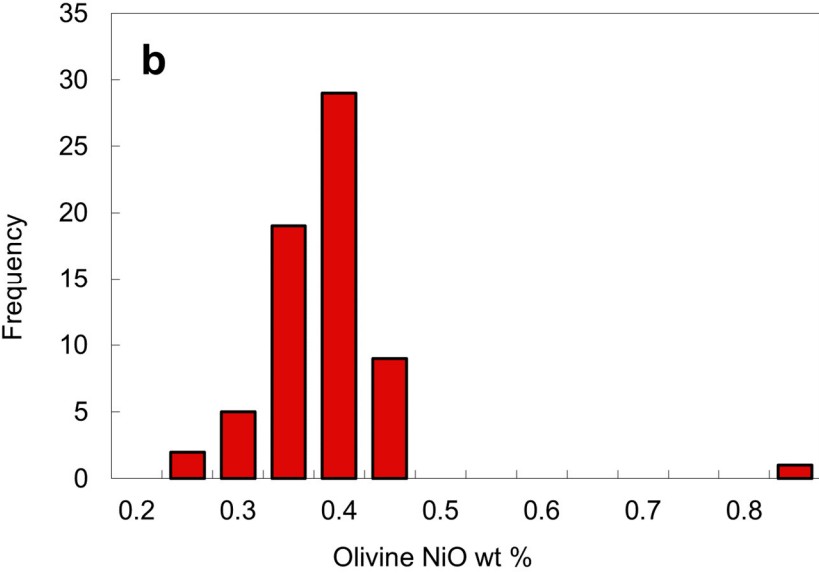

**Figure 9.** Histograms of olivine inclusions in diamond from the No. 50 kimberlite diatreme: (**a**) Mg # defined as $100Mg/(Mg + Fe^{2+})$ by atom, where all Fe in olivine is assumed to $Fe^{2+}$; (**b**) NiO contents. The data are from Table 2.

**Table 2.** Average compositions of olivine inclusions in diamonds from the No. 50 kimberlite diatreme.

| Sample | LN50D01 | LN50D02 | LN50D02 | LN50D03 | LN50D03 | LN50D03 | LN50D04 | LN50D04 | LN50D06 | LN50D06 | LN50D06 | LN50D11 | LN50D11 |
|---|---|---|---|---|---|---|---|---|---|---|---|---|---|
| Inclusion | | 1 (large) | 2 (small) | 1 (small) | 2 (large) | 3 | 1 | 2 | 1 | 2 | 3 | 1 | 2 |
| $SiO_2$ | 41.21 | 39.98 | 40.61 | 41.08 | 41.36 | 41.22 | 41.38 | 42.02 | 41.20 | 40.69 | 40.69 | 40.58 | 41.14 |
| $TiO_2$ | 0.01 | 0.00 | 0.00 | 0.00 | 0.00 | 0.01 | 0.01 | 0.00 | 0.01 | 0.00 | 0.01 | 0.01 | 0.00 |
| $Al_2O_3$ | 0.01 | 0.01 | 0.02 | 0.01 | 0.01 | 0.02 | 0.01 | 0.01 | 0.00 | 0.01 | 0.02 | 0.00 | 0.01 |
| $Cr_2O_3$ | 0.05 | 0.04 | 0.04 | 0.08 | 0.08 | 0.13 | 0.07 | 0.09 | 0.07 | 0.06 | 0.04 | 0.04 | 0.02 |
| TFeO | 7.39 | 7.33 | 7.11 | 6.38 | 6.52 | 6.35 | 6.48 | 6.39 | 7.08 | 7.24 | 7.05 | 7.49 | 7.56 |
| NiO | 0.37 | 0.35 | 0.34 | 0.34 | 0.35 | 0.39 | 0.34 | 0.26 | 0.37 | 0.36 | 0.34 | 0.39 | 0.38 |
| MnO | 0.11 | 0.10 | 0.10 | 0.09 | 0.13 | 0.06 | 0.11 | 0.09 | 0.09 | 0.12 | 0.12 | 0.11 | 0.09 |
| MgO | 51.28 | 51.32 | 51.17 | 51.62 | 51.74 | 51.14 | 51.27 | 51.72 | 51.35 | 51.02 | 50.98 | 50.72 | 50.18 |
| CaO | 0.02 | 0.03 | 0.04 | 0.02 | 0.03 | 0.02 | 0.02 | 0.03 | 0.03 | 0.03 | 0.04 | 0.04 | 0.04 |
| $Na_2O$ | na | na | na | na | na | 0.02 | 0.00 | 0.00 | na | na | na | na | 0.02 |
| Σ | 100.45 | 99.17 | 99.42 | 99.62 | 100.22 | 99.37 | 99.69 | 100.61 | 100.20 | 99.53 | 99.29 | 99.37 | 99.44 |
| Si | 0.994 | 0.975 | 0.988 | 0.995 | 0.997 | 1.003 | 1.004 | 1.009 | 0.996 | 0.990 | 0.992 | 0.990 | 1.005 |
| Al | 0.000 | 0.000 | 0.000 | 0.000 | 0.000 | 0.001 | 0.000 | 0.000 | 0.000 | 0.000 | 0.001 | 0.000 | 0.000 |
| Ti | 0.000 | 0.000 | 0.000 | 0.000 | 0.000 | 0.000 | 0.000 | 0.000 | 0.000 | 0.000 | 0.000 | 0.000 | 0.000 |
| Cr | 0.001 | 0.001 | 0.001 | 0.002 | 0.001 | 0.002 | 0.001 | 0.002 | 0.001 | 0.001 | 0.001 | 0.001 | 0.000 |
| Fe | 0.149 | 0.149 | 0.145 | 0.129 | 0.131 | 0.129 | 0.132 | 0.128 | 0.143 | 0.147 | 0.144 | 0.153 | 0.154 |
| Ni | 0.007 | 0.007 | 0.007 | 0.007 | 0.007 | 0.008 | 0.007 | 0.005 | 0.007 | 0.007 | 0.007 | 0.008 | 0.007 |
| Mn | 0.002 | 0.002 | 0.002 | 0.002 | 0.003 | 0.001 | 0.002 | 0.002 | 0.002 | 0.002 | 0.002 | 0.002 | 0.002 |
| Mg | 1.845 | 1.865 | 1.856 | 1.865 | 1.859 | 1.854 | 1.854 | 1.852 | 1.850 | 1.851 | 1.853 | 1.845 | 1.828 |
| Ca | 0.001 | 0.001 | 0.001 | 0.000 | 0.001 | 0.001 | 0.000 | 0.001 | 0.001 | 0.001 | 0.001 | 0.001 | 0.001 |
| Na | 0.000 | 0.000 | 0.000 | 0.000 | 0.000 | 0.001 | 0.000 | 0.000 | 0.000 | 0.000 | 0.000 | 0.000 | 0.001 |
| Σcation | 3.000 | 3.000 | 3.000 | 3.000 | 3.000 | 3.000 | 3.000 | 3.000 | 3.000 | 3.000 | 3.000 | 3.000 | 3.000 |
| ΣO | 3.995 | 3.975 | 3.989 | 3.996 | 3.998 | 4.004 | 4.005 | 4.010 | 3.997 | 3.991 | 3.993 | 3.991 | 4.005 |
| Mg # | 92.5 | 92.6 | 92.8 | 93.5 | 93.4 | 93.5 | 93.4 | 93.5 | 92.8 | 92.6 | 92.8 | 92.4 | 92.2 |

Mg # = $100Mg/(Fe + Mg)$ by atoms; na: not analyzed; $Fe^{3+}$ not calculated.

**Table 2.** (continued) Average compositions of olivine inclusions in diamonds from the No. 50 kimberlite diatreme.

| | LN50D12 | LN50D12 | LN50D14 | LN50D14 | LN50D20 | LN50D21 | LN50D30 | LN50D35 | LN50D35 | LN50D39 | LN50D44 | LN50D45 | LN50D50 | LN50D50 |
|---|---|---|---|---|---|---|---|---|---|---|---|---|---|---|
| | 1 (large) | 2 (small) | 1 | 2 | | | | 1 (large) | 2 (small) | | | | 1 (large) | 2 |
| | 41.52 | 40.92 | 40.94 | 40.48 | 40.81 | 40.83 | 39.71 | 41.13 | 41.18 | 41.34 | 41.54 | 41.11 | 40.76 | 40.95 |
| | 0.02 | 0.00 | 0.00 | 0.01 | 0.00 | 0.01 | 0.01 | 0.01 | 0.00 | 0.00 | 0.01 | 0.00 | 0.01 | 0.00 |
| | 0.05 | 0.12 | 0.04 | 0.34 | 0.02 | 0.02 | 0.02 | 0.01 | 0.01 | 0.02 | 0.00 | 0.01 | 0.01 | 0.02 |
| | 0.08 | 0.07 | 0.08 | 0.05 | 0.02 | 0.08 | 0.04 | 0.04 | 0.07 | 0.08 | 0.10 | 0.06 | 0.05 | 0.07 |
| | 7.78 | 7.56 | 7.70 | 7.79 | 7.73 | 7.05 | 8.07 | 6.87 | 6.77 | 7.33 | 7.59 | 6.97 | 7.61 | 7.92 |
| | 0.36 | 0.43 | 0.42 | 0.44 | 0.40 | 0.35 | 0.42 | 0.35 | 0.30 | 0.38 | 0.39 | 0.45 | 0.41 | 0.40 |
| | 0.11 | 0.13 | 0.06 | 0.12 | 0.11 | 0.09 | 0.13 | 0.09 | 0.10 | 0.15 | 0.10 | 0.12 | 0.12 | 0.05 |
| | 50.78 | 49.89 | 50.68 | 50.41 | 50.90 | 51.20 | 49.15 | 51.62 | 52.10 | 50.45 | 50.86 | 51.73 | 50.80 | 50.35 |
| | 0.02 | 0.03 | 0.04 | 0.09 | 0.01 | 0.03 | 0.05 | 0.02 | 0.02 | 0.04 | 0.04 | 0.03 | 0.03 | 0.04 |
| | 0.02 | 0.01 | 0.01 | 0.01 | na | 0.01 | 0.02 | 0.01 | 0.01 | na | na | 0.01 | 0.01 | 0.02 |
| | 100.74 | 99.14 | 99.97 | 99.73 | 100.01 | 99.67 | 97.62 | 100.15 | 100.55 | 99.78 | 100.63 | 100.48 | 99.82 | 99.82 |
| | 1.002 | 1.004 | 0.995 | 0.986 | 0.990 | 0.991 | 0.990 | 0.993 | 0.989 | 1.006 | 1.003 | 0.989 | 0.991 | 0.997 |
| | 0.001 | 0.003 | 0.001 | 0.010 | 0.000 | 0.001 | 0.001 | 0.000 | 0.000 | 0.001 | 0.000 | 0.000 | 0.000 | 0.000 |
| | 0.000 | 0.000 | 0.000 | 0.000 | 0.000 | 0.000 | 0.000 | 0.000 | 0.000 | 0.000 | 0.000 | 0.000 | 0.000 | 0.000 |
| | 0.002 | 0.001 | 0.002 | 0.001 | 0.000 | 0.002 | 0.001 | 0.001 | 0.001 | 0.001 | 0.002 | 0.001 | 0.001 | 0.001 |
| | 0.157 | 0.155 | 0.156 | 0.159 | 0.157 | 0.143 | 0.168 | 0.139 | 0.136 | 0.149 | 0.153 | 0.140 | 0.155 | 0.161 |
| | 0.007 | 0.008 | 0.008 | 0.009 | 0.008 | 0.007 | 0.008 | 0.007 | 0.006 | 0.007 | 0.008 | 0.009 | 0.008 | 0.008 |
| | 0.002 | 0.003 | 0.001 | 0.002 | 0.002 | 0.002 | 0.003 | 0.002 | 0.002 | 0.003 | 0.002 | 0.002 | 0.002 | 0.001 |
| | 1.827 | 1.825 | 1.835 | 1.831 | 1.842 | 1.853 | 1.827 | 1.858 | 1.865 | 1.831 | 1.831 | 1.856 | 1.841 | 1.828 |
| | 0.001 | 0.001 | 0.001 | 0.002 | 0.000 | 0.001 | 0.001 | 0.000 | 0.000 | 0.001 | 0.001 | 0.001 | 0.001 | 0.001 |
| | 0.001 | 0.000 | 0.000 | 0.000 | 0.000 | 0.000 | 0.001 | 0.001 | 0.000 | 0.000 | 0.000 | 0.001 | 0.000 | 0.001 |
| | 3.000 | 3.000 | 3.000 | 3.000 | 3.000 | 3.000 | 3.000 | 3.000 | 3.000 | 3.000 | 3.000 | 3.000 | 3.000 | 3.000 |
| | 4.003 | 4.006 | 3.996 | 3.991 | 3.991 | 3.993 | 3.990 | 3.993 | 3.990 | 4.007 | 4.004 | 3.990 | 3.992 | 3.998 |
| | 92.1 | 92.2 | 92.1 | 92.0 | 92.1 | 92.8 | 91.6 | 93.1 | 93.2 | 92.5 | 92.3 | 93.0 | 92.3 | 91.9 |

**Table 2.** (continued) Average compositions of olivine inclusions in diamonds from the No. 50 kimberlite diatreme.

| LN50D69 1 (large) | LN50D69 2 (small) | LN50D69 | LN50D70 | LN50D72 | LN50D74 | LN50D79 | LN50D91 1 (large) | LN50D91 2 (small) | LN50D91 3 | LN50D91 4 | LN50D94 | LN50D95 | LN50D96 | LN50D99 |
|---|---|---|---|---|---|---|---|---|---|---|---|---|---|---|
| 40.54 | 41.21 | 41.06 | 41.39 | 41.83 | 40.75 | 40.87 | 41.70 | 40.82 | 40.76 | 40.98 | 41.29 | 41.27 | 40.19 | 41.01 |
| 0.01 | 0.00 | 0.00 | 0.01 | 0.00 | 0.02 | 0.01 | 0.01 | 0.01 | 0.00 | 0.00 | 0.01 | 0.00 | 0.01 | 0.00 |
| 0.02 | 0.03 | 0.02 | 0.01 | 0.05 | 0.04 | 0.00 | 0.01 | 0.01 | 0.01 | 0.01 | 0.01 | 0.01 | 0.02 | 0.00 |
| 0.10 | 0.07 | 0.06 | 0.06 | 0.08 | 0.06 | 0.07 | 0.03 | 0.06 | 0.05 | 0.03 | 0.03 | 0.04 | 0.03 | 0.04 |
| 6.82 | 6.65 | 6.67 | 6.53 | 7.29 | 6.75 | 6.94 | 7.68 | 7.61 | 7.75 | 7.65 | 6.85 | 6.72 | 7.29 | 7.08 |
| 0.34 | 0.36 | 0.35 | 0.31 | 0.36 | 0.29 | 0.35 | 0.36 | 0.36 | 0.33 | 0.34 | 0.36 | 0.32 | 0.33 | 0.40 |
| 0.12 | 0.07 | 0.06 | 0.06 | 0.14 | 0.12 | 0.10 | 0.11 | 0.09 | 0.06 | 0.08 | 0.11 | 0.10 | 0.07 | 0.04 |
| 50.78 | 50.33 | 51.82 | 50.85 | 50.94 | 51.12 | 50.92 | 51.39 | 50.84 | 51.13 | 51.27 | 51.94 | 51.55 | 50.73 | 52.02 |
| 0.06 | 0.03 | 0.05 | 0.02 | 0.03 | 0.02 | 0.03 | 0.03 | 0.02 | 0.02 | 0.02 | 0.04 | 0.02 | 0.04 | 0.01 |
| 0.00 | 0.02 | 0.03 | 0.01 | 0.01 | 0.00 | 0.01 | 0.01 | 0.00 | 0.01 | 0.01 | 0.00 | 0.00 | 0.01 | 0.01 |
| 98.78 | 98.76 | 100.12 | 99.24 | 100.74 | 99.17 | 99.30 | 101.32 | 99.81 | 100.12 | 100.38 | 100.64 | 100.05 | 98.72 | 100.61 |
| 0.993 | 1.011 | 0.990 | 1.009 | 1.008 | 0.993 | 0.996 | 0.999 | 0.992 | 0.987 | 0.990 | 0.992 | 0.997 | 0.985 | 0.985 |
| 0.000 | 0.001 | 0.001 | 0.000 | 0.002 | 0.001 | 0.000 | 0.000 | 0.000 | 0.000 | 0.000 | 0.000 | 0.000 | 0.001 | 0.000 |
| 0.000 | 0.000 | 0.000 | 0.000 | 0.000 | 0.000 | 0.000 | 0.000 | 0.000 | 0.000 | 0.000 | 0.000 | 0.000 | 0.000 | 0.000 |
| 0.002 | 0.001 | 0.001 | 0.001 | 0.002 | 0.001 | 0.001 | 0.001 | 0.001 | 0.001 | 0.001 | 0.001 | 0.001 | 0.001 | 0.001 |
| 0.140 | 0.136 | 0.134 | 0.133 | 0.147 | 0.138 | 0.141 | 0.154 | 0.155 | 0.157 | 0.154 | 0.138 | 0.136 | 0.149 | 0.142 |
| 0.007 | 0.007 | 0.007 | 0.006 | 0.007 | 0.006 | 0.007 | 0.007 | 0.007 | 0.006 | 0.007 | 0.007 | 0.006 | 0.007 | 0.008 |
| 0.002 | 0.001 | 0.001 | 0.001 | 0.003 | 0.002 | 0.002 | 0.002 | 0.002 | 0.001 | 0.002 | 0.002 | 0.002 | 0.002 | 0.001 |
| 1.854 | 1.841 | 1.863 | 1.848 | 1.830 | 1.858 | 1.850 | 1.836 | 1.842 | 1.846 | 1.846 | 1.859 | 1.857 | 1.854 | 1.863 |
| 0.002 | 0.001 | 0.001 | 0.001 | 0.001 | 0.001 | 0.001 | 0.001 | 0.000 | 0.001 | 0.001 | 0.001 | 0.001 | 0.001 | 0.000 |
| 0.000 | 0.001 | 0.002 | 0.000 | 0.001 | 0.000 | 0.000 | 0.001 | 0.000 | 0.000 | 0.000 | 0.000 | 0.000 | 0.000 | 0.000 |
| 3.000 | 3.000 | 3.000 | 3.000 | 3.000 | 3.000 | 3.000 | 3.000 | 3.000 | 3.000 | 3.000 | 3.000 | 3.000 | 3.000 | 3.000 |
| 3.994 | 4.011 | 3.990 | 4.010 | 4.009 | 3.995 | 3.997 | 3.999 | 3.993 | 3.988 | 3.990 | 3.992 | 3.998 | 3.986 | 3.985 |
| 93.0 | 93.1 | 93.3 | 93.3 | 92.6 | 93.1 | 92.9 | 92.3 | 92.3 | 92.2 | 92.3 | 93.1 | 93.2 | 92.5 | 92.9 |

Chromium contents of the olivine inclusions (Table 2; Figure 10a) all fall within the normal reported range of olivine inclusions in diamonds [24] and appear to increase with Mg #, likely owing to different degree of depletion in the mantle. If an olivine inclusion touches or is close to a chromite inclusion in diamond, an apparent high $Cr_2O_3$ content of the olivine may be caused by secondary X-ray fluorescence of Cr in chromite by Fe in olivine [25,26]. Therefore, a low accelerating voltage (10 kV) is preferred for analysis of olivine inclusions in olivine-chromite pairs in diamond. The unusually high Cr content of olivine inclusions in the Akwatia diamond [24] should be checked for fluorescence effect. Nonetheless, high Cr contents in isolated olivine inclusions are likely be real. Chromium enrichment in olivine might result from specific P-T conditions, crystal-chemical factors, or reduction of Cr, which may enter olivine as $Cr^{2+}$ under reduced conditions [36] or high crystallization temperatures [37,38]. Sutton et al. [39] showed that Cr is predominantly divalent in lunar olivine.

Chromium contents of the olivine inclusions (Table 2; Figure 10a) all fall within the normal reported range of olivine inclusions in diamonds [24] and appear to increase with Mg #, probably due to different degree of depletion in the mantle. If an olivine inclusion touches or is close to a chromite inclusion in diamond, an apparent high $Cr_2O_3$ content of the olivine may be caused by secondary X-ray fluorescence of Cr in chromite by Fe in olivine [26]. Therefore, a low accelerating voltage (10 kV) is preferred for analysis of olivine inclusions in olivine-chromite pairs in diamond. The unusually high Cr content of olivine inclusions in the Akwatia diamond [24] should be checked for fluorescence effect. Nonetheless, high Cr contents in isolated olivine inclusions are likely be real. Chromium enrichment in olivine might result from specific P-T conditions, crystal-chemical factors, or reduction in Cr, which may enter olivine as $Cr^{2+}$ under reduced conditions [36] or high crystallization temperatures [37,38]. Sutton et al. [39] showed that Cr is predominantly divalent in lunar olivine.

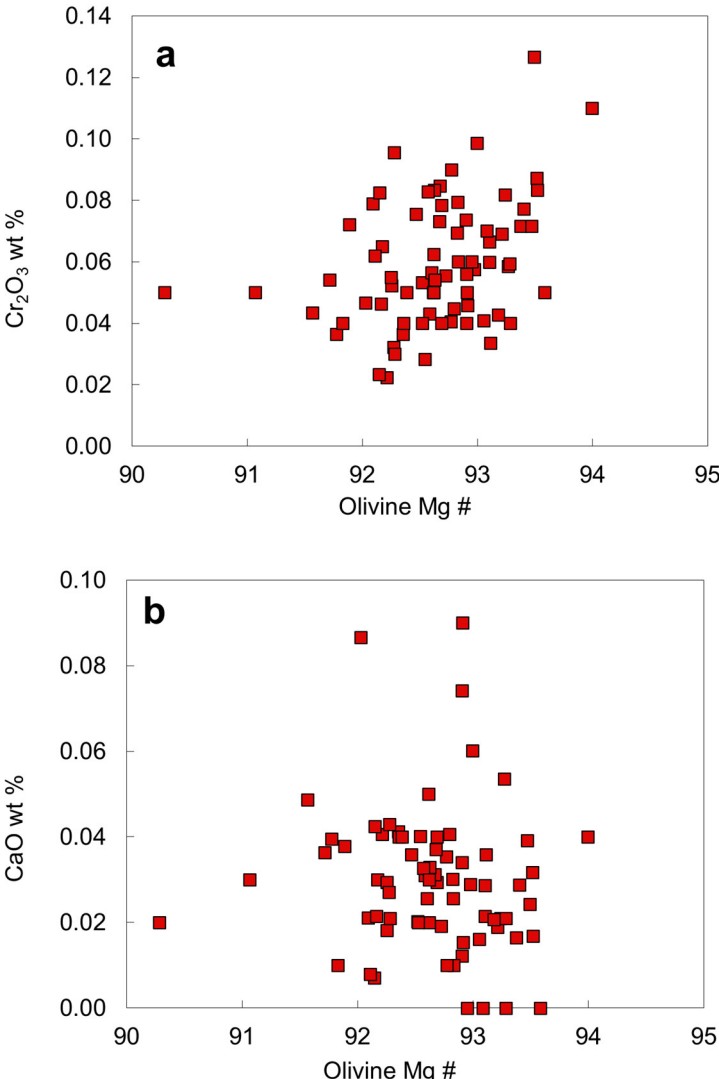

**Figure 10.** Compositional plots for the olivine inclusions in diamond: (**a**) Mg # versus Cr2O3; (**b**) Mg # versus CaO.

The Ca content of olivine is an indicator of pressure if it is in equilibrium with clinopyroxene. This is the basis of Ca exchange barometer between olivine and clinopyroxene [40], although it is a system that depends much more on temperature than on pressure [41]. The advantage of applying this system to olivine in diamond is that it is less likely to be reset during late stage processes. Very few olivine inclusions from North China contain CaO exceeding 0.06 wt. % (Table 2; Figure 10b), whereas olivine inclusions in diamond from other localities have CaO mainly in the range 0.00–0.13 and ≤0.23 wt. % [24]. As indicated by the experimental data on Ca exchange between olivine and clinopyroxene [40], a low Ca content in olivine (<0.08 wt. %) would require a high pressure, consistent with diamond stability field. The low Ca content of olivine inclusions might imply a Ca depletion of their source region (i.e., a harzburgite source) or crystallization at a lower temperature or a higher pressure than usual (if from lherzolite).

**Chromite.** Thirteen chromite inclusions were analyzed (Table 3) and additional chromite analyses from the Fuxian pipes are available in the literature [16–18] . Chromite occurs either as separate inclusions or with other minerals in a single diamond. Diamond commonly contains both chromite and olivine inclusions. The Mg # of the chromite inclusions varies from 60 to 70 with the peak position at a Mg # of 66 (Figure 11), overlapping those of the Fe-rich part of the peridotitic worldwide range (Mg # of 60–80 [24]). Recalculated $100Fe^{2+}/(Fe^{3+} + Fe^{2+})$ ratios are relatively high (76–95), probably indicating a relatively

reduced environment. The $Al_2O_3$ ranges from 4.0 to 7.9 wt. %, within the worldwide range of chromite inclusions in diamond [24]. A striking feature of the chromite inclusions in diamond is a high Cr content (63–67 wt. % $Cr_2O_3$ Table 3), which lie within the field defined by worldwide chromite inclusions in diamond [1], but are centered around 66 wt. % $Cr_2O_3$ (Figure 11).

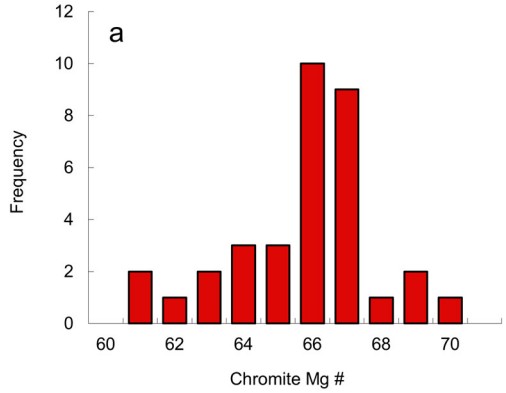

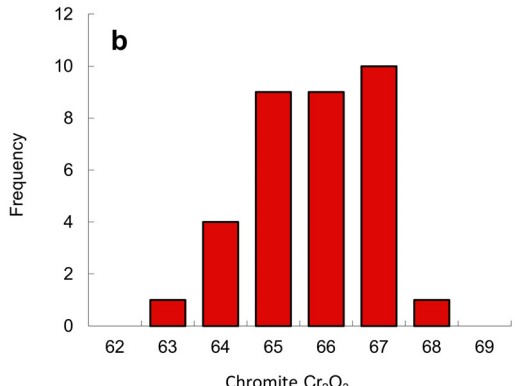

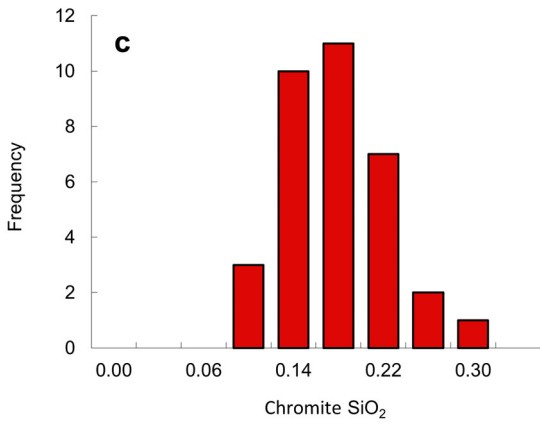

**Figure 11.** Histograms of chromite inclusions in diamond: (**a**) Mg #; (**b**) $Cr_2O_3$; (**c**) $SiO_2$. The data are listed in Table 3.

**Table 3.** Compositions of chromite inclusions in diamonds from the No. 50 kimberlite diatreme.

| Sample | LN50D03 | LN50D04 | | | | LN50D07 | | | | LN50D12 | LN50D20 | | LN50D58 | |
|---|---|---|---|---|---|---|---|---|---|---|---|---|---|---|
| Grain | | Sp2 | Sp 3 | Sp 4 | Sp 5 | Sp 1 | Sp 2 | Sp 3 | Sp 4 | Sp | Sp 1 | Sp 2 | Sp 1 | Sp 2 |
| $SiO_2$ | 0.20 | 0.18 | 0.16 | 0.19 | 0.19 | 0.12 | 0.12 | 0.11 | 0.15 | 0.17 | 0.14 | 0.14 | 0.16 | 0.21 |
| $TiO_2$ | 0.20 | 0.22 | 0.19 | 0.18 | 0.17 | 0.41 | 0.32 | 0.42 | 0.41 | 0.45 | 0.05 | 0.06 | 0.03 | 0.04 |
| $Al_2O_3$ | 6.28 | 5.83 | 6.08 | 5.77 | 5.88 | 4.83 | 4.82 | 4.82 | 4.73 | 4.20 | 4.92 | 4.96 | 6.72 | 4.76 |
| $Cr_2O_3$ | 64.80 | 66.76 | 66.48 | 65.93 | 65.34 | 66.10 | 66.31 | 66.28 | 66.74 | 66.72 | 64.26 | 65.00 | 64.31 | 66.90 |
| $V_2O_3$ | 0.18 | 0.14 | 0.16 | 0.17 | 0.14 | 0.16 | 0.17 | 0.18 | 0.22 | 0.30 | 0.25 | 0.26 | 0.14 | 0.19 |
| TFeO | 13.38 | 13.41 | 13.25 | 13.32 | 13.24 | 15.41 | 15.39 | 15.16 | 15.03 | 15.79 | 16.42 | 16.62 | 13.76 | 14.22 |
| NiO | 0.00 | 0.09 | 0.15 | 0.10 | 0.13 | 0.04 | 0.13 | 0.09 | 0.14 | 0.16 | 0.08 | 0.09 | 0.15 | 0.00 |
| MnO | 0.00 | 0.23 | 0.00 | 0.00 | 0.00 | 0.00 | 0.00 | 0.00 | 0.00 | 0.00 | 0.24 | 0.32 | 0.00 | 0.00 |
| MgO | 14.22 | 13.74 | 14.23 | 13.28 | 14.43 | 12.74 | 12.72 | 13.46 | 13.43 | 12.56 | 11.96 | 12.08 | 13.70 | 13.37 |
| ZnO | 0.10 | 0.04 | 0.07 | 0.07 | 0.01 | 0.05 | 0.03 | 0.09 | 0.06 | 0.08 | 0.09 | 0.08 | 0.06 | 0.03 |
| Σ | 99.36 | 100.64 | 100.77 | 99.01 | 99.55 | 99.86 | 100.00 | 100.60 | 100.90 | 100.43 | 98.41 | 99.61 | 99.03 | 99.71 |
| $Fe_2O_3$ | 2.10 | 1.26 | 1.71 | 0.78 | 2.41 | 1.84 | 1.88 | 2.57 | 2.27 | 1.89 | 2.78 | 2.84 | 2.02 | 1.80 |
| FeO | 11.49 | 12.28 | 11.71 | 12.62 | 11.08 | 13.76 | 13.70 | 12.84 | 12.99 | 14.09 | 13.92 | 14.07 | 11.94 | 12.60 |
| Σ | 99.57 | 100.77 | 100.94 | 99.09 | 99.79 | 100.05 | 100.19 | 100.86 | 101.13 | 100.62 | 98.69 | 99.90 | 99.24 | 99.90 |
| Si | 0.006 | 0.006 | 0.005 | 0.006 | 0.006 | 0.004 | 0.004 | 0.004 | 0.005 | 0.006 | 0.005 | 0.005 | 0.005 | 0.007 |
| Alvi | 0.242 | 0.224 | 0.232 | 0.226 | 0.227 | 0.189 | 0.188 | 0.186 | 0.182 | 0.164 | 0.196 | 0.195 | 0.261 | 0.185 |
| $Fe^{+3}$ | 0.052 | 0.031 | 0.042 | 0.019 | 0.059 | 0.046 | 0.047 | 0.063 | 0.056 | 0.047 | 0.071 | 0.071 | 0.050 | 0.045 |
| Ti | 0.005 | 0.005 | 0.005 | 0.005 | 0.004 | 0.010 | 0.008 | 0.010 | 0.010 | 0.011 | 0.001 | 0.002 | 0.001 | 0.001 |
| Cr | 1.678 | 1.719 | 1.702 | 1.729 | 1.690 | 1.733 | 1.737 | 1.717 | 1.726 | 1.747 | 1.715 | 1.714 | 1.673 | 1.749 |
| V | 0.005 | 0.004 | 0.004 | 0.004 | 0.004 | 0.004 | 0.005 | 0.005 | 0.006 | 0.008 | 0.007 | 0.007 | 0.004 | 0.005 |
| $Fe^{+2}$ | 0.315 | 0.335 | 0.317 | 0.350 | 0.303 | 0.382 | 0.380 | 0.352 | 0.355 | 0.390 | 0.393 | 0.392 | 0.329 | 0.348 |
| Ni | 0.000 | 0.002 | 0.004 | 0.003 | 0.003 | 0.001 | 0.003 | 0.002 | 0.004 | 0.004 | 0.002 | 0.002 | 0.004 | 0.000 |
| Mn | 0.000 | 0.006 | 0.000 | 0.000 | 0.000 | 0.000 | 0.000 | 0.000 | 0.000 | 0.000 | 0.007 | 0.009 | 0.000 | 0.000 |
| Mg | 0.694 | 0.667 | 0.687 | 0.657 | 0.704 | 0.630 | 0.628 | 0.658 | 0.655 | 0.620 | 0.602 | 0.601 | 0.672 | 0.659 |
| Zn | 0.002 | 0.001 | 0.002 | 0.002 | 0.000 | 0.001 | 0.001 | 0.002 | 0.001 | 0.002 | 0.002 | 0.002 | 0.001 | 0.001 |
| Σcation | 3.000 | 3.000 | 3.000 | 3.000 | 3.000 | 3.000 | 3.000 | 3.000 | 3.000 | 3.000 | 3.000 | 3.000 | 3.000 | 3.000 |
| ΣO | 4.000 | 4.000 | 4.000 | 4.000 | 4.000 | 4.000 | 4.000 | 4.000 | 4.000 | 4.000 | 4.000 | 4.000 | 4.000 | 4.000 |
| Mg # | 68.8 | 66.6 | 68.4 | 65.2 | 69.9 | 62.3 | 62.3 | 65.1 | 64.8 | 61.4 | 60.5 | 60.5 | 67.2 | 65.4 |
| Cr # | 87.4 | 88.5 | 88.0 | 88.5 | 88.2 | 90.2 | 90.2 | 90.2 | 90.4 | 91.4 | 89.8 | 89.8 | 86.5 | 90.4 |

Mg # = $100Mg/(Fe^{2+} + Mg)$ and Cr # = $100\ Cr/(Cr + Al)$ by atom; ferric iron by charge balance and stoichiometry.

Except for two inclusions with 0.79 and 0.75 wt. % $TiO_2$, the $TiO_2$ contents (0.03–0.45 wt. %) of most chromites falls within the compositional range defined by worldwide chromites (maximum 0.65 wt. %), located in the "diamond inclusion field" for chromites (Figure 12a). On a $Fe_2O_3$ vs. $Al_2O_3$ diagram, analyses of chromite inclusions fall within a limited compositional range (Figure 12b). The recalculated $Fe_2O_3$ of the chromite inclusions is less than 4.5 wt. %. The $SiO_2$ contents of chromite inclusions are high, from 0.08 to 0.29 wt. %, with a pronounced peak around 0.18 wt. % (Figure 11c). The solution of silicate-spinel component is favored by high pressure [42], consistent with the high $Cr_2O_3$ contents that stabilize spinel towards higher pressures in lherzolites [43].

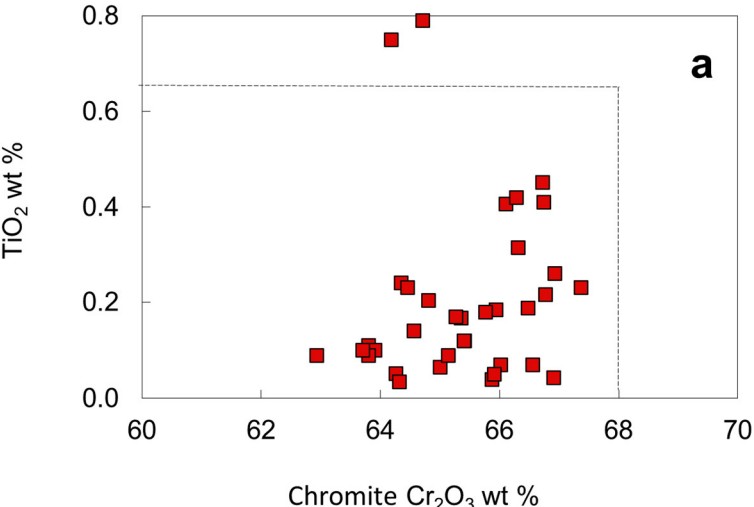

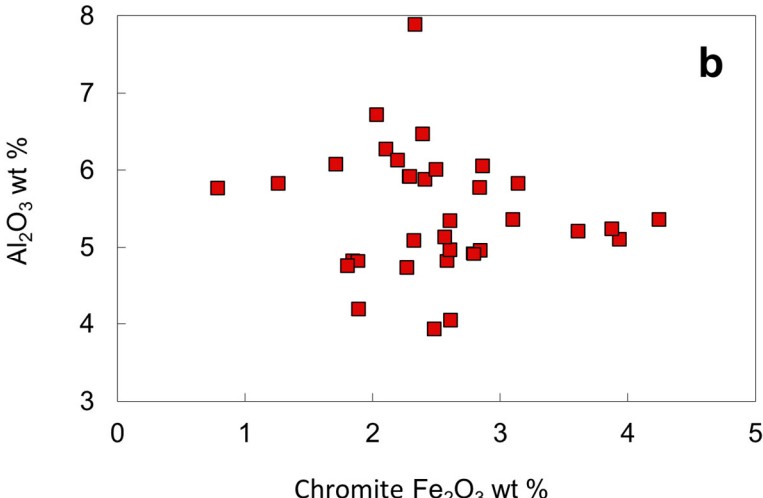

**Figure 12.** Compositional plots for chromite inclusions in diamond from the No. 50 kimberlite: (**a**) Cr2O3 versus TiO2; (**b**) Fe2O3 versus Al2O3. Dashed lines delineate a "diamond inclusion field" [23]. The total Fe from EPMA is calculated into $Fe^{2+}$ and $Fe^{3+}$.

Some diamonds contain more than one separated chromite inclusion (e.g., samples LN50D04, LN50D07 and LN50D20). Sample LN50D04 has 6 chromite inclusions, some at the edge of the diamond polished section, and others near the center of the diamond host; some are large and euhedral, while others are small and anhedral. There are no significant variations in composition for the multiple spinel grains in the diamond, although such inter-grain compositional variations have been recorded previously [22,23,44,45]. Multiple chromites in a single diamond have nearly the same compositions, probably indicating that the diamond formed in a very short time, or that all the chromite inclusions formed in the same environment, or that the composition of chromite did not evolve during growth of the diamond.

**Garnet.** Five garnet inclusions in diamond were exposed by polishing and analyzed (Table 4). Garnet occurs either as a single inclusion or together with other mineral phases. Eclogitic garnet is extremely rare. No undisputed eclogitic inclusion was discovered in

this study. A garnet numbered 52A in a garnet-orthopyroxene-clinopyroxene assemblage reported by Harris et al. [16] is closest to eclogitic almandine-pyrope. Garnets in diamond from the No. 50 diatreme have the following: a Mg # of 81–88 (except for sample 52A that coexists with a low magnesian orthopyroxene and clinopyroxene); a Cr # of 19–39 except for samples 43A and 52A, where Cr # is defined as 100Cr/(Cr + Al) by atom; a Ca # of 5–21, where the Ca # is defined as 100Ca/(Ca + Mg) by atom (Figure 13).

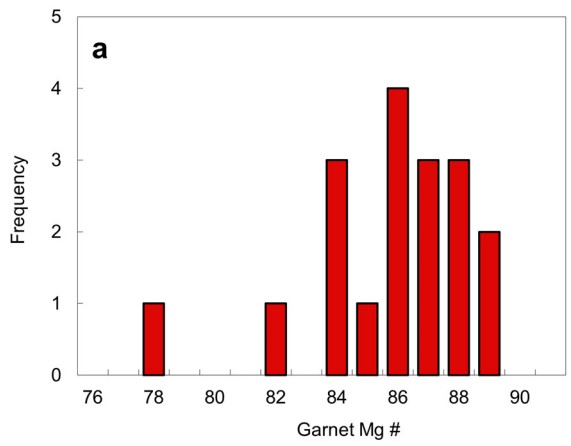

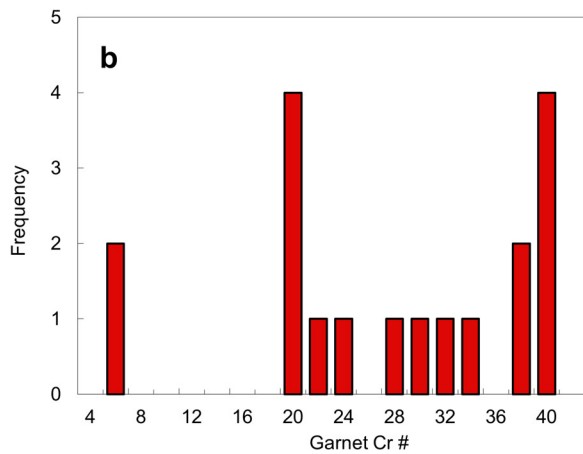

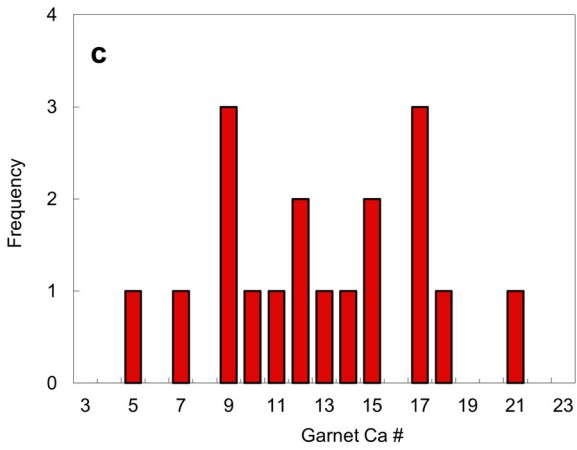

**Figure 13.** Histograms of garnet inclusions in diamond: (**a**) Mg #; (**b**) Cr # defined as 100Cr/(Cr + Al) by atom; (**c**) Ca # = 100Ca/(Ca + Mg) by atom. The data are from Table 4.

**Table 4.** Compositions of garnet inclusions in diamond from the No. 50 kimberlite diatreme.

| Sample | LN50D10 | LN50D10 | LN50D13 | LN50D19 | LN50D68 | LN50D71 | 40A | 43A | 52A | WG1 | WG5 | WG6 | WG7 | WG8 | L1 | L2 | L3 | L4 |
|---|---|---|---|---|---|---|---|---|---|---|---|---|---|---|---|---|---|---|
| $SiO_2$ | 41.83 | 42.38 | 41.22 | 40.78 | 41.05 | 40.52 | 42.83 | 42.21 | 41.24 | 41.49 | 41.70 | 41.00 | 41.34 | 40.22 | 41.50 | 40.60 | 41.20 | 42.00 |
| $TiO_2$ | 0.04 | 0.03 | 0.07 | 0.05 | 0.05 | 0.09 | 0.02 | 0.05 | 0.41 | 0.02 | 0.01 | 0.02 | 1.08 | 0.00 | 0.01 | 0.03 | 0.00 | 0.00 |
| $Al_2O_3$ | 19.19 | 19.13 | 15.29 | 14.64 | 18.06 | 15.34 | 18.56 | 21.94 | 22.27 | 18.84 | 13.50 | 13.50 | 15.26 | 15.23 | 14.40 | 13.60 | 16.40 | 17.30 |
| $Cr_2O_3$ | 6.66 | 6.78 | 11.29 | 13.59 | 7.46 | 13.38 | 6.74 | 1.76 | 1.48 | 6.61 | 12.80 | 12.70 | 8.42 | 10.37 | 12.10 | 12.70 | 9.90 | 8.10 |
| TFeO | 5.43 | 5.46 | 6.66 | 6.15 | 6.69 | 5.88 | 6.08 | 7.66 | 10.36 | 6.99 | 6.52 | 6.52 | 7.75 | 6.23 | 5.95 | 5.65 | 6.19 | 6.99 |
| NiO | 0.03 | 0.03 | 0.05 | 0.00 | 0.05 | 0.03 | 0.00 | 0.00 | 0.00 | 0.00 | 0.00 | 0.00 | na | na | 0.00 | 0.02 | 0.01 | 0.07 |
| MnO | 0.22 | 0.17 | 0.22 | 0.15 | 0.24 | 0.21 | 0.36 | 0.32 | 0.38 | 0.39 | 0.35 | 0.36 | 0.15 | 0.21 | 0.30 | 0.32 | 0.26 | 0.38 |
| MgO | 23.11 | 22.65 | 22.04 | 21.59 | 20.52 | 21.64 | 23.52 | 21.04 | 19.77 | 20.01 | 21.60 | 21.00 | 19.05 | 20.80 | 21.70 | 22.00 | 23.70 | 20.30 |
| CaO | 2.91 | 3.01 | 3.25 | 3.51 | 5.83 | 2.79 | 2.30 | 4.85 | 4.31 | 5.96 | 4.23 | 4.80 | 6.71 | 5.68 | 3.79 | 3.94 | 1.67 | 5.66 |
| $Na_2O$ | 0.00 | 0.01 | 0.01 | 0.00 | 0.01 | 0.03 | 0.01 | 0.04 | 0.07 | 0.02 | 0.00 | 0.00 | 0.07 | na | 0.00 | 0.02 | 0.01 | 0.04 |
| Σ | 99.42 | 99.64 | 100.10 | 100.46 | 99.97 | 99.92 | 100.42 | 99.87 | 100.29 | 100.33 | 100.71 | 99.90 | 99.83 | 98.74 | 99.75 | 98.88 | 99.34 | 100.84 |
| $Fe_2O_3$ | 0.65 | 0.00 | 1.19 | 0.55 | 2.18 | 0.00 | 0.15 | 1.52 | 2.39 | 1.28 | 1.03 | 1.58 | 0.42 | 3.01 | 0.18 | 1.97 | 1.81 | 0.94 |
| FeO | 4.85 | 5.46 | 5.59 | 5.66 | 4.74 | 5.88 | 5.95 | 6.29 | 8.21 | 5.84 | 5.59 | 5.10 | 7.37 | 3.53 | 5.79 | 3.87 | 4.56 | 6.15 |
| Σ | 99.49 | 99.64 | 100.21 | 100.51 | 100.18 | 99.92 | 100.43 | 100.02 | 100.53 | 100.46 | 100.81 | 100.06 | 99.87 | 99.04 | 99.77 | 99.08 | 99.52 | 100.93 |
| Si | 2.986 | 3.026 | 2.988 | 2.965 | 2.960 | 2.956 | 3.033 | 2.993 | 2.941 | 2.980 | 3.027 | 3.004 | 3.031 | 2.956 | 3.027 | 2.986 | 2.972 | 3.016 |
| Aliv | 0.014 | 0.000 | 0.012 | 0.035 | 0.040 | 0.044 | 0.000 | 0.007 | 0.059 | 0.020 | 0.000 | 0.000 | 0.000 | 0.044 | 0.000 | 0.014 | 0.028 | 0.000 |
| Alvi | 1.601 | 1.610 | 1.294 | 1.219 | 1.494 | 1.274 | 1.549 | 1.827 | 1.813 | 1.575 | 1.155 | 1.166 | 1.318 | 1.275 | 1.238 | 1.165 | 1.366 | 1.464 |
| $Fe^{+3}$ | 0.035 | 0.000 | 0.065 | 0.030 | 0.118 | 0.000 | 0.008 | 0.081 | 0.128 | 0.069 | 0.056 | 0.087 | 0.023 | 0.166 | 0.010 | 0.109 | 0.098 | 0.051 |
| Ti | 0.002 | 0.002 | 0.004 | 0.003 | 0.003 | 0.005 | 0.001 | 0.003 | 0.022 | 0.001 | 0.001 | 0.001 | 0.060 | 0.000 | 0.001 | 0.002 | 0.000 | 0.000 |
| Cr | 0.376 | 0.383 | 0.647 | 0.781 | 0.425 | 0.772 | 0.377 | 0.099 | 0.083 | 0.375 | 0.735 | 0.736 | 0.488 | 0.603 | 0.698 | 0.739 | 0.565 | 0.460 |
| $Fe^{+2}$ | 0.290 | 0.326 | 0.339 | 0.344 | 0.286 | 0.358 | 0.352 | 0.373 | 0.490 | 0.351 | 0.340 | 0.312 | 0.452 | 0.217 | 0.353 | 0.238 | 0.275 | 0.369 |
| Ni | 0.002 | 0.002 | 0.003 | 0.000 | 0.003 | 0.002 | 0.000 | 0.000 | 0.000 | 0.000 | 0.000 | 0.000 | 0.000 | 0.000 | 0.000 | 0.001 | 0.001 | 0.004 |
| Mn | 0.013 | 0.010 | 0.013 | 0.009 | 0.014 | 0.013 | 0.022 | 0.019 | 0.023 | 0.024 | 0.022 | 0.022 | 0.009 | 0.013 | 0.019 | 0.020 | 0.016 | 0.023 |
| Mg | 2.459 | 2.410 | 2.382 | 2.340 | 2.205 | 2.353 | 2.483 | 2.224 | 2.102 | 2.143 | 2.337 | 2.294 | 2.082 | 2.279 | 2.359 | 2.412 | 2.549 | 2.173 |
| Ca | 0.223 | 0.230 | 0.252 | 0.273 | 0.450 | 0.218 | 0.174 | 0.368 | 0.329 | 0.459 | 0.329 | 0.377 | 0.527 | 0.447 | 0.296 | 0.311 | 0.129 | 0.435 |
| Na | 0.001 | 0.001 | 0.001 | 0.000 | 0.002 | 0.005 | 0.001 | 0.005 | 0.010 | 0.003 | 0.000 | 0.000 | 0.010 | 0.000 | 0.000 | 0.003 | 0.001 | 0.006 |
| Σcation | 8.000 | 8.000 | 8.000 | 8.000 | 8.000 | 8.000 | 8.000 | 8.000 | 8.000 | 8.000 | 8.000 | 8.000 | 8.000 | 8.000 | 8.000 | 8.000 | 8.000 | 8.000 |
| ΣO | 12.000 | 12.024 | 12.000 | 12.000 | 12.000 | 12.003 | 12.000 | 12.000 | 12.000 | 12.000 | 12.000 | 12.000 | 12.000 | 12.000 | 12.000 | 12.000 | 12.000 | 12.000 |
| Mg # | 88.3 | 88.1 | 85.5 | 86.2 | 84.5 | 86.8 | 87.3 | 83.0 | 77.3 | 83.6 | 85.5 | 85.2 | 81.4 | 85.6 | 86.7 | 87.4 | 87.2 | 83.8 |
| Ca # | 8.3 | 8.7 | 9.6 | 10.5 | 17.0 | 8.5 | 6.6 | 14.2 | 13.5 | 17.6 | 12.3 | 14.1 | 20.2 | 16.4 | 11.2 | 11.4 | 4.8 | 16.7 |
| Cr # | 18.9 | 19.2 | 33.1 | 38.4 | 21.7 | 36.9 | 19.6 | 5.1 | 4.3 | 19.1 | 38.9 | 38.7 | 27.0 | 31.4 | 36.0 | 38.5 | 28.8 | 23.9 |

Ca # = 100Ca/(Ca + Mg) by atom; na: not analyzed; $Fe^{3+}$ by charge balance and stoichiometry. Data from this study unless otherwise indicated. 40A, 43A and 52A: Harris et al. [16]. 40A and 43A also reported by Wang and Guo [18]. WG1, WG5–WG8: Wang and Guo [18]. L1–L4: Meyer et al. [17].

To distinguish diamondiferous from non-diamondiferous assemblages on the basis of the composition of peridotitic pyrope, curves on a CaO versus $Cr_2O_3$ diagram (Figure 14) were derived by various authors [27,46,47] to separate a high (lherzolitic) and a low (harzburgitic) calcium field. Most garnets in diamond from the No. 50 diatreme belong to the harzburgitic paragenesis, and only a few garnet-bearing diamond crystals are part of the lherzolitic association. Chemical compositions of two garnets (samples 43A and 52A, cf. [16]) plot below the dashed line distinguishing peridotitic from non-peridotitic garnets [47], indicating that they may belong to a different paragenesis. The garnet 52A has the lowest Mg # and it also coexists with low Mg # orthopyroxene and clinopyroxene, and therefore the assemblage garnet-orthopyroxene-clinopyroxene may belongs to a websterite or eclogitic paragenesis.

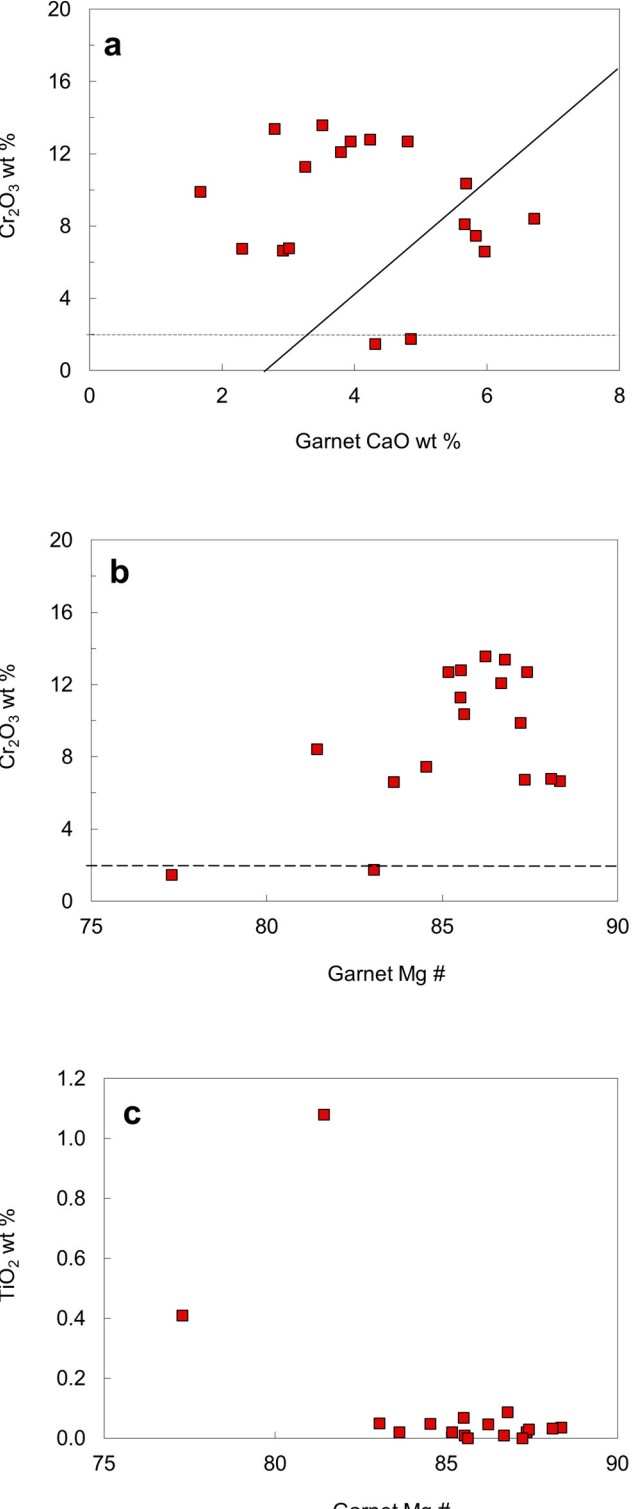

**Figure 14.** Compositional plots for garnet inclusions in diamond: (**a**) CaO versus Cr2O3; (**b**) Mg # versus Cr2O3; (**c**) Mg # versus TiO2. Solid line divides compositional fields for lherzolitic and harzburgitic garnets, dashed line distinguishes peridotitic from non-peridotitic garnets. Both lines are described in detail by Gurney and Zweistra [47].

Figure 14c shows that only lherzolitic garnets with the lowest Mg # contain significant amounts of TiO2 (0.41 and 1.08 wt. %), while the other garnets contain less than 0.09 wt. % TiO2. The Al/Cr ratio in garnet was taken as a measure for the fertility of the source,

as there is a positive correlation between Al/Cr ratio and Ti in lherzolitic garnets [24]. However, there is no positive correlation between Al/Cr ratio and Ti content for the garnet inclusions from the No. 50 diamond association.

Most garnet inclusions have Si atoms per formula unit (apfu) close to the ideal value of 3 when normalized to 12 oxygens. A few garnets have Si slightly greater than 3, up to 3.03 apfu (Table 4). The slight excess of Si in garnet may suggest the presence of majoritic component ($Mg_3(MgSi)Si_3O_{12}$) [32] and further suggests a high-pressure condition if equilibrated with orthopyroxene [48–50]. The diamond LN50D10 has two garnet inclusions. One is small (about 80 μm across) and has no excess Si. The other, exposed after polishing away the first garnet, has a small amount of excess Si (approximately 1%). These two garnets have almost identical compositions except for Si. The difference in Si might be caused by systematic errors between different EPMA sessions. Since one garnet was no longer available, it was not possible to examine systematic errors between different EPMA sessions by analyzing both garnets during a same EPMA session. Therefore, the garnet inclusion with excess Si was re-analyzed, together with a garnet xenocryst sample without excess Si (a Four Corner ultramafic garnet, PY15 [51]). The average of the second garnet inclusion from the second EPMA session has less excess Si (3.003 apfu) relative to the first EPMA session. However, the average of garnet xenocrysts from the same EPMA session also has less Si (2.977 apfu) relative to early EPMA analyses (2.994 apfu). Although inconclusive, these measurements suggest that the small excess Si is real, probably indicating the existence of a very minor majoritic component.

**Pyroxene.** Four orthopyroxene inclusions in three diamond hosts were recovered (Table 5). In addition, Harris et al. [16] reported two coexisting orthopyroxene inclusions and one websteritic orthopyroxene inclusion. Clinopyroxene was not identified in this work, although six clinopyroxene inclusions were recovered by Harris et al. [16] . Two coexisting orthopyroxene grains in the diamond LN50D40 are homogeneous and have the same composition (Table 5). The Mg # of the orthopyroxene is about 94, except for the websteritic orthopyroxene with a Mg # of 88 (sample 52B, cf. [16]).

**Table 5.** Compositions of pyroxene inclusions in diamond.

| | Orthopyroxene | | | | | | Clinopyroxene | | | |
|---|---|---|---|---|---|---|---|---|---|---|
| Sample | LN50D40 | LN50D40 | LN50D52 | LN50D65 | 38A | 52B | Sample | 28B | 43B | 52B |
| Grain | 1 | 2 | | | | | Grain | | | |
| $SiO_2$ | 58.24 | 57.75 | 57.69 | 57.39 | 58.62 | 56.63 | $SiO_2$ | 55.25 | 55.41 | 54.48 |
| $TiO_2$ | 0.00 | 0.02 | 0.01 | 0.02 | 0.00 | 0.11 | $TiO_2$ | 0.07 | 0.01 | 0.19 |
| $Al_2O_3$ | 0.43 | 0.62 | 0.95 | 0.56 | 0.51 | 0.76 | $Al_2O_3$ | 0.76 | 1.23 | 2.35 |
| $Cr_2O_3$ | 0.53 | 0.53 | 0.49 | 0.48 | 0.51 | 0.13 | $Cr_2O_3$ | 1.51 | 0.36 | 0.64 |
| TFeO | 4.30 | 4.33 | 4.26 | 4.59 | 4.18 | 7.47 | TFeO | 1.98 | 2.52 | 3.95 |
| NiO | 0.16 | 0.10 | 0.13 | 0.10 | 0.11 | 0.00 | NiO | 0.00 | 0.00 | 0.00 |
| MnO | 0.15 | 0.12 | 0.14 | 0.11 | 0.14 | 0.13 | MnO | 0.09 | 0.07 | 0.14 |
| MgO | 35.22 | 35.08 | 36.41 | 35.49 | 35.62 | 32.22 | MgO | 18.40 | 18.23 | 16.97 |
| CaO | 0.42 | 0.42 | 0.33 | 0.74 | 0.26 | 2.29 | CaO | 20.89 | 21.48 | 18.75 |
| $Na_2O$ | 0.11 | 0.12 | 0.03 | 0.14 | 0.00 | 0.31 | $Na_2O$ | 0.84 | 0.77 | 1.81 |
| Σ | 99.57 | 99.09 | 100.44 | 99.63 | 99.95 | 100.05 | Σ | 99.79 | 100.08 | 99.28 |
| Si | 1.002 | 0.998 | 0.981 | 0.986 | 1.005 | 0.984 | Si | 0.999 | 0.998 | 0.989 |
| Aliv | 0.000 | 0.002 | 0.019 | 0.011 | 0.000 | 0.016 | Aliv | 0.001 | 0.002 | 0.011 |
| Alvi | 0.009 | 0.011 | 0.000 | 0.000 | 0.010 | 0.000 | Alvi | 0.016 | 0.024 | 0.039 |
| Ti | 0.000 | 0.000 | 0.000 | 0.000 | 0.000 | 0.001 | Ti | 0.001 | 0.000 | 0.003 |
| Cr | 0.007 | 0.007 | 0.007 | 0.007 | 0.007 | 0.002 | Cr | 0.022 | 0.005 | 0.009 |
| Fe | 0.062 | 0.063 | 0.061 | 0.066 | 0.060 | 0.108 | Fe | 0.030 | 0.038 | 0.060 |
| Ni | 0.002 | 0.001 | 0.002 | 0.001 | 0.002 | 0.000 | Ni | 0.000 | 0.000 | 0.000 |
| Mn | 0.002 | 0.002 | 0.002 | 0.002 | 0.002 | 0.002 | Mn | 0.001 | 0.001 | 0.002 |
| Mg | 0.904 | 0.904 | 0.923 | 0.909 | 0.910 | 0.834 | Mg | 0.496 | 0.490 | 0.459 |
| Ca | 0.008 | 0.008 | 0.006 | 0.014 | 0.005 | 0.043 | Ca | 0.405 | 0.415 | 0.365 |
| Na | 0.004 | 0.004 | 0.001 | 0.005 | 0.000 | 0.010 | Na | 0.029 | 0.027 | 0.064 |

| Σcation | 2.000 | 2.000 | 2.000 | 2.000 | 2.000 | 2.000 | ∑cation | 2.000 | 2.000 | 2.000 |
| ΣO | 3.009 | 3.006 | 2.993 | 2.993 | 3.013 | 2.988 | ∑O | 3.005 | 3.001 | 2.989 |
| Mg # | 93.6 | 93.5 | 93.8 | 93.2 | 93.8 | 88.5 | Mg# | 94.3 | 92.8 | 88.5 |
| | | | | | | | Ca# | 44.9 | 45.9 | 44.3 |

Mg# = 100Mg/(Fe + Mg) by atoms; Ca# = 100Ca/(Ca + Mg) by atoms; $Fe^{3+}$ not calculated. 38A, 52B, 28B and 43B from Harris et al. [16] and Wang and Guo [18]. 38A: two co-existing opx with the same composition. 52B: websteritic, co-existing clinopyroxene, orthopyroxene and garnet. 28B contains K2O 0.22 wt % and co-exists with olivine. 43B contains K2O 0.07 wt % and co-exists with garnet.

The orthopyroxene inclusions have low Ca contents (CaO < 0.50 wt. % except for the websteritic and the green orthopyroxene in LN50D65), probably indicating that some orthopyroxene might not be in equilibrium with clinopyroxene and might represent a harzburgitic origin [52]. Compared to harzburgitic orthopyroxene, the websteritic orthopyroxene has higher Ti, Al, Fe, Ca and Na and lower Si, Cr and Mg (Table 5; Figure 15).

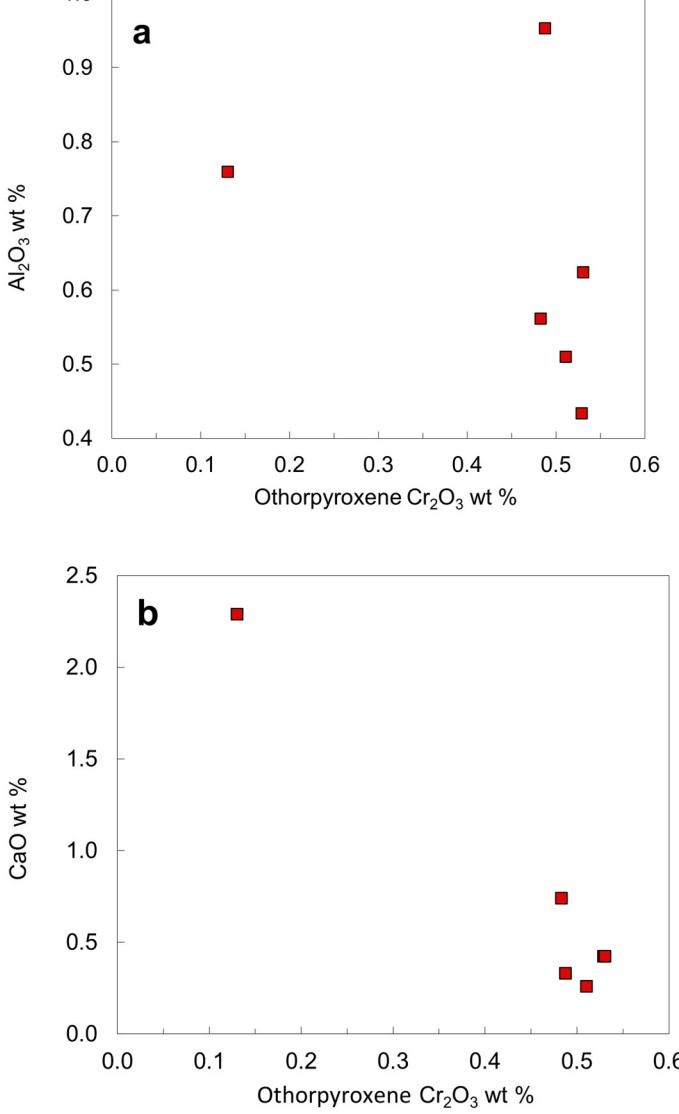

**Figure 15.** Compositional plots for orthopyroxene inclusions in diamond from the No. 50 kimberlite diatreme: (**a**) Cr2O3 versus Al2O3; (**b**) Cr2O3 versus CaO. All the harzburgitic orthopyroxenes are similar, but different from the websteritic orthopyroxene.

The olivine and orthopyroxene inclusions in the No 50 diamond are slightly enriched in iron compared with harzburgitic phases from worldwide sources [1], probably

suggesting a relatively fertile character of harzburgitic mantle. In agreement with the expected olivine–orthopyroxene partitioning relationship in peridotite xenoliths [53], the peak of Mg # for the orthopyroxene is slightly higher than that of the olivine.

### 6.5. Chemical Compositions of Rare Mineral Inclusions

**Carbonate.** Carbonate inclusions in diamond are rare. The Ca-carbonate inclusion in sample LN50D11 (Figure 6b) was initially described as a silicate under optical microscope. The analyses using a EPMA procedure for silicate showed that the Ca-carbonate inclusion is mainly composed of $CaCO_3$ with minor $MgCO_3$ and $FeCO_3$. The loss of the Ca-carbonate inclusion precludes further study of this interesting sample. A magnesite inclusion was found in a fracture of the diamond LN50D13 and coexists with a garnet grain. The coexistence of magnesite with garnet can be used to obtain an upper limit of oxygen fugacity from the reaction $6MgCO_3 + 22Al_2SiO_5 + 4SiO_2 = 2Mg_3Al_2Si_3O_{12} + 6C + 9O_2$ and a lower limit of oxygen fugacity in absence of periclase from the reaction $MgCO_3 = MgO + C + O_2$. Wang et al. [54] reported a magnesite inclusion in a diamond from the No. 50 kimberlite. Carbonate inclusions in diamonds from other kimberlites include Ca carbonate, magnesite, and dolomite [55–58].

**Sulfides.** Four primary inclusions and one secondary sulfide inclusion in diamond were exposed and analyzed. Representative chemical compositions of the sulfide inclusions are given in Table 6. Harris et al. [16] reported pyrrhotite with less than 1.0 wt. % of Co, Ni, Cu and Zn and pentlandite with 34–35 wt. % Ni in diamonds from the No. 50 kimberlite, but did not provide complete analyses. Some sulfide inclusions from other localities are given in Table 6 for comparison.

**Table 6.** Representative compositions of sulfide inclusions in diamonds from the No. 50 kimberlite diatreme.

| Sample | LN50D04 | | | | | LN50D32 | | LN50D37 | | LN50D42 | LN50D70 |
|---|---|---|---|---|---|---|---|---|---|---|---|
| Mineral | grg | grg | trl | Cu po | pnt | R₃S₄ | | R₃S₄ | | po | py |
| Fe | 34.13 | 55.66 | 31.10 | 29.81 | 36.55 | 37.35 | 34.43 | 36.55 | 36.49 | 29.54 | 43.21 |
| Ni | 16.73 | 2.84 | 33.08 | 9.78 | 26.19 | 13.91 | 17.33 | 19.22 | 18.13 | 33.75 | 0.80 |
| Cu | 8.64 | 0.06 | 0.06 | 22.00 | 0.15 | 0.00 | 4.65 | 0.00 | 0.76 | 0.33 | 0.00 |
| Zn | 0.00 | 0.00 | 0.00 | 0.00 | 0.00 | 0.00 | 0.00 | 0.00 | 0.01 | 0.00 | 0.00 |
| Mn | 0.00 | 0.01 | 0.00 | 0.00 | 0.00 | 0.00 | 0.00 | 0.02 | 0.00 | 0.01 | 0.01 |
| Co | 0.31 | 0.00 | 0.42 | 0.01 | 0.30 | 0.78 | 0.52 | 0.67 | 0.81 | 0.34 | 0.80 |
| Cr | na | na | na | na | 0.74 | 0.27 | 0.37 | 0.13 | 0.65 | na | na |
| S | 40.47 | 41.90 | 36.18 | 36.54 | 34.36 | 49.08 | 43.18 | 42.22 | 43.22 | 36.23 | 57.17 |
| Σ | 100.29 | 100.46 | 100.83 | 98.15 | 98.29 | 101.39 | 100.47 | 98.81 | 100.07 | 100.18 | 102.00 |
| Sulfur normalized to 4 | | | | | | | | | | | |
| Fe | 1.936 | 3.051 | 1.974 | 1.873 | 2.442 | 1.747 | 1.831 | 1.988 | 1.938 | 1.872 | 1.736 |
| Ni | 0.903 | 0.148 | 1.998 | 0.585 | 1.665 | 0.619 | 0.877 | 0.995 | 0.916 | 2.035 | 0.030 |
| Cu | 0.431 | 0.003 | 0.003 | 1.215 | 0.009 | 0.000 | 0.217 | 0.000 | 0.035 | 0.018 | 0.000 |
| Zn | 0.000 | 0.000 | 0.000 | 0.000 | 0.000 | 0.000 | 0.000 | 0.000 | 0.000 | 0.000 | 0.000 |
| Mn | 0.000 | 0.000 | 0.000 | 0.000 | 0.000 | 0.000 | 0.000 | 0.001 | 0.000 | 0.000 | 0.000 |
| Co | 0.017 | 0.000 | 0.025 | 0.001 | 0.019 | 0.035 | 0.026 | 0.034 | 0.041 | 0.020 | 0.030 |
| Cr | 0.000 | 0.000 | 0.000 | 0.000 | 0.053 | 0.013 | 0.021 | 0.008 | 0.037 | 0.000 | 0.000 |
| S | 4.000 | 4.000 | 4.000 | 4.000 | 4.000 | 4.000 | 4.000 | 4.000 | 4.000 | 4.000 | 4.000 |
| Σcation | 3.287 | 3.202 | 4.000 | 3.674 | 4.189 | 2.415 | 2.972 | 3.025 | 2.969 | 3.947 | 1.797 |

grg: greigite; trl: troilite; po: pyrrhotite; pnt: pentlandite; vlr: violarite; py: pyrite. na: not analyzed.

A sulfide inclusion in the diamond LN50D04 has a heterogeneous composition and shows Fe-rich, Ni-rich, and Cu-rich domains on X-ray map (Figure 7). The content of Cu could be high as 22 wt. % in some areas. The Σcation/sulfur ratio of the most sulfide analyses is <1 (down to 0.800), indicating that the sulfide is predominantly pyrrhotite ($R_{1-x}S$). One analysis with Σcation/sulfur = 1 may be troilite and one with a ratio of 1.047 may be

pentlandite. Normalized to four atoms, some analyses have $\Sigma$cations slightly higher than, although close to, four, suggesting the existence of $R_3S_4$ minerals in the linnaeite group that includes violarite $FeNi_2S_4$, daubreelite $FeCr_2S_4$, greigite $Fe_3S_4$, and carrollite $Cu(Co,Ni)_2S_4$. The sulfide inclusion in LN50D04 was initially analyzed at 20 kV and 10 nA. At these conditions, the Fe counts of Ni-rich domains could increase due to the fluorescence of Fe in the adjacent Fe-rich domain by Ni in the Ni-rich domain, thus producing excess cations relative to $R_3S_4$ phases. To examine this possibility, the inclusion was analyzed again at 10 kV and 20 nA condition. The effects of secondary fluorescence were not detected, and the calculated $\Sigma$cation/sulfur ratios remain similar. Chromium contents of the sulfide inclusion may be up to 0.85 wt. % (Figure 16d); K, Mn, Co, As, Se, Sb, Te, Ba and Bi in the sulfide are low or below the EPMA detection limit.

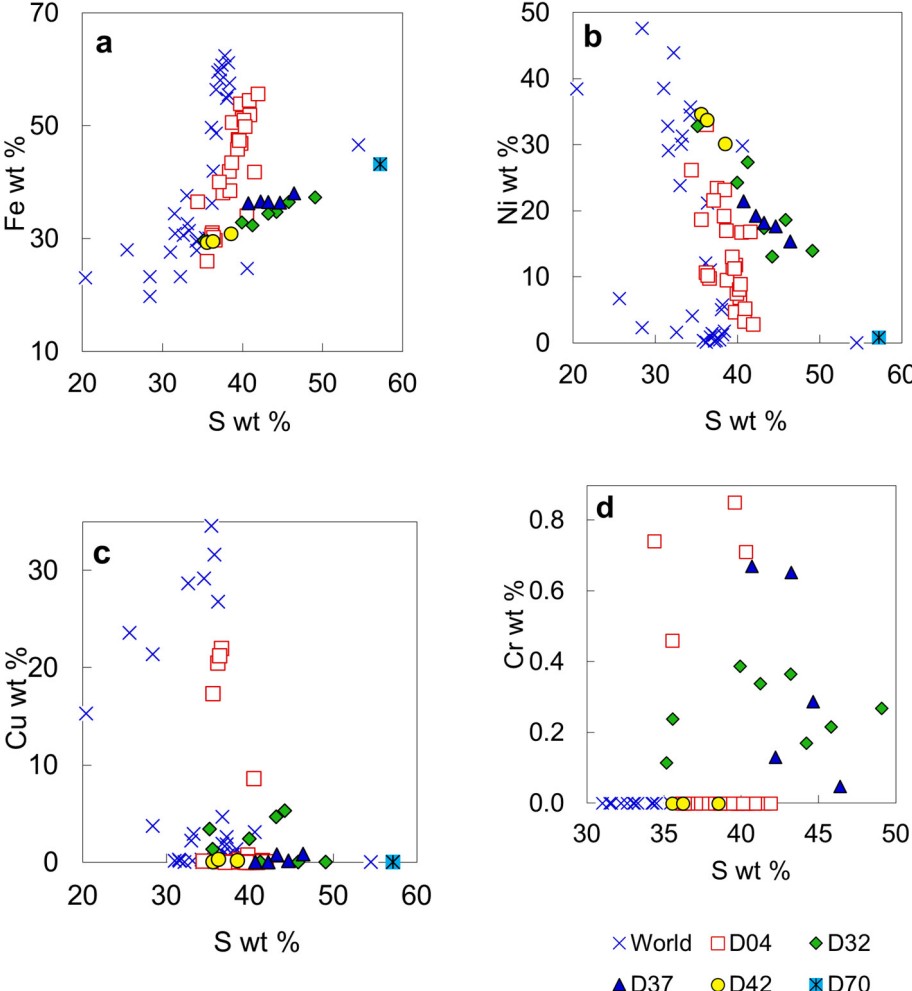

**Figure 16.** Compositional plots for sulfide inclusions in diamond from the No. 50 kimberlite diatreme: (**a**) S versus Fe; (**b**) S versus Ni; (**c**) S versus Cu; (**d**) S versus Cr.

Sulfides from LN50D32 and LN50D37 have formulae near $NiFe_2S_4$ (Table 6). The sulfide in LN50D37 contains no Cu and its chemical variations are insignificant. This sulfide appears to be a solid solution between greigite ($Fe^{2+}Fe^{3+}_2S_4$) with polydymite ($Ni^{2+}Ni^{3+}_2S_4$), and/or violarite ($Fe^{2+}Ni^{3+}_2S_4$). The exact identity depends on knowing the valence of Ni and Fe, which could vary during the P-T history of the sulfide. Moreover, the composition of the LN50D37 sulfide is likely an average of a submicroscopic intergrowth. Sulfide from the LN50D42 belongs to pyrrhotite or pentlandite groups. Sulfide from the LN50D70 is located in a fracture and is secondary in origin (Table 6).

Bulanova et al. [59] used a proton microprobe to analyze sulfide inclusions in diamond. However, sulfide inclusions are compositionally heterogeneous and their sizes are usually small (mostly 20–50 μm in this work) and similar to the typical size of the proton beam spot on the sample (30–50 μm). Therefore, the large volumes (essentially entire inclusions) analyzed by the proton microprobe will give information on the average bulk composition of the original sulfide or melt.

**Unknown hydrous silicate phase.** The unknown silicate phase in sample LN50D67 (Figure 5c, 5d) is yellowish optically and has no fractures connected to the diamond surface. The inclusion has two compositional domains with one domain enriched in MgO and poor in $SiO_2$ relative to the other. The EPMA totals are low, from 85.9 to 90.2 wt. %, as is the case for serpentine, chlorite and humite (Table 7). The inclusion contains ~10 wt. % $Al_2O_3$, higher than most serpentine and lower than most chlorite, common alteration products of olivine. If $H_2O$ is the only other component, the inclusion is a hydrous silicate in the system $MgO-FeO-Al_2O_3-SiO_2-H_2O$. There is another hydrous silicate inclusion reported in diamond from the same locality (Table 7). Hydrous dense silicate phases in system $MgO-SiO_2-H_2O$ could be stable at very high pressure [60–62]. The unknown hydrous silicate phase is a hydrous Mg-Fe aluminosilicate, similar to Phase F ($1.2MgO \cdot 1.8SiO_2 \cdot 1.2H_2O$) in MgO and $H_2O$ if Al substitutes for Si and Mg simultaneously, or close to Phase D ($MgO \cdot SiO_2 \cdot H_2O$) or Phase E ($2.3MgO \cdot 1.25SiO_2 \cdot 1.2H_2O$) in $SiO_2$ and $H_2O$. The sample was lost during additional polishing, and no additional characterization could be undertaken.

**Table 7.** Compositions of unknown hydrous phases in diamonds from the No. 50 kimberlite diatreme.

| | LN50D67 Domain 1 | LN50D67 Domain 2 | Miao et al. [10] | | Smyth & Kawamoto [61] | | Burnley and Navrotsky [60] | | |
|---|---|---|---|---|---|---|---|---|---|
| **Mineral** | | | | | **Wadsleyite II** | | **Phase D** | **Phase E** | **Phase F** |
| **Average** | **3 Analyses** | **3 Analyses** | **4 Analyses** | | **1** | **2** | | | |
| $SiO_2$ | 47.27 | 41.29 | 65.81 | $SiO_2$ | 40.04 | 40.01 | 50.74 | 39.65 | 60.71 |
| $TiO_2$ | 0.01 | 0.01 | 0.06 | $TiO_2$ | na | na | 0.00 | 0.00 | 0.00 |
| $Al_2O_3$ | 10.01 | 10.42 | 0.91 | $Al_2O_3$ | 0.41 | 0.34 | 0.00 | 0.00 | 0.00 |
| $Cr_2O_3$ | 0.08 | 0.10 | 0.11 | $Cr_2O_3$ | 0.19 | 0.22 | 0.00 | 0.00 | 0.00 |
| FeO | 8.19 | 11.90 | 0.22 | FeO | 8.80 | 10.60 | 0.00 | 0.00 | 0.00 |
| NiO | 0.13 | 0.21 | na | NiO | na | na | 0.00 | 0.00 | 0.00 |
| MnO | 0.01 | 0.02 | 0.02 | MnO | na | na | 0.00 | 0.00 | 0.00 |
| MgO | 20.37 | 24.39 | 29.36 | MgO | 47.57 | 44.37 | 34.04 | 48.94 | 27.15 |
| CaO | 0.87 | 0.38 | 0.11 | CaO | 0.00 | 0.00 | 0.00 | 0.00 | 0.00 |
| Σ | 86.95 | 88.72 | 96.60 | $H_2O$ | 2.99 | 4.46 | 15.21 | 11.41 | 12.14 |
| $H_2O$ | 12.90 | 12.68 | 4.84 | Σ | 100.00 | 100.00 | 100.00 | 100.00 | 100.00 |
| Σ | 99.85 | 101.40 | 101.44 | | | | | | |
| Si | 4.395 | 3.906 | 4.08 | Si | 0.975 | 0.980 | 1.000 | 1.250 | 1.800 |
| Al | 1.097 | 1.162 | 0.07 | Al | 0.012 | 0.010 | 0.000 | 0.000 | 0.000 |
| Ti | 0.001 | 0.000 | 0.00 | Ti | 0.000 | 0.000 | 0.000 | 0.000 | 0.000 |
| Cr | 0.006 | 0.008 | 0.01 | Cr | 0.004 | 0.004 | 0.000 | 0.000 | 0.000 |
| Fe | 0.637 | 0.942 | 0.01 | Fe | 0.179 | 0.217 | 0.000 | 0.000 | 0.000 |
| Ni | 0.008 | 0.013 | 0.00 | Ni | 0.000 | 0.000 | 0.000 | 0.000 | 0.000 |
| Mn | 0.000 | 0.001 | 0.00 | Mn | 0.000 | 0.000 | 0.000 | 0.000 | 0.000 |
| Mg | 2.822 | 3.439 | 2.71 | Mg | 1.726 | 1.620 | 1.000 | 2.300 | 1.200 |
| Ca | 0.087 | 0.039 | 0.01 | Ca | 0.000 | 0.000 | 0.000 | 0.000 | 0.000 |
| Σcation | 9.053 | 9.509 | 6.88 | H | 0.243 | 0.364 | 2.000 | 2.400 | 2.400 |
| ΣO | 14.000 | 14.000 | 11.00 | Σcation | 3.139 | 3.195 | 4.000 | 5.950 | 5.400 |
| Mg # | 81.6 | 78.5 | 99.58 | ΣO | 4.000 | 4.000 | 4.000 | 6.000 | 6.000 |

na: not analyzed. LN50D67: based on $6MgO.4SiO2.4H2O$, normalized to 14 O excluding O in H2O. Samples from Miao et al. [10]: based on $3MgO.4SiO2.H2O$, normalized to 11 O excluding O in H2O.

**Fe-rich phase.** An Fe-rich phase in the diamond LN50D09 was discovered and analyzed for Fe, Co, Ni, Cu, Mn, Ba, Al, Si, and Cr, and the total oxides are approximately 80 wt. %, indicating the presence of other components (Table 8). Thus, the phase is neither native iron, wüstite, nor magnetite. Assuming the additional species is $H_2O$, the phase is likely to be a hydrous iron oxide, such as goethite, FeOOH (Table 8). Other elements detected in the phase are Si (up to 3.8 wt. % $SiO_2$) and Al (up to 0.37 wt. % $Al_2O_3$) (Table 8). The paragenesis of this Fe-rich inclusion is unknown. The diamond host of this inclusion is cloudy and of poor quality. Although there is no visible fracture observed in the diamond host, the possibility of penetration of external components into the inside of diamond cannot be completely excluded. The inclusion was probably originally trapped as native iron, wüstite, or magnetite, and was later altered or modified by an external fluid. A similar Fe-dominant inclusion in diamond was described by Miao et al. [10] from the same locality (Table 8). Significant $SiO_2$ (0.8 wt. %) was also reported in a Fe-phase by Stachel et al. [57].

**Table 8.** Compositions of an unknown Fe-rich phase in diamond from the No. 50 kimberlite diatreme.

| Analysis | LN50D09 (Goethite or Limonite?) | | | | | | | Miao et al. [10] | |
| | a1 | a2 | b1 | b2 | b3 | b4 | Average | Goethite (?) | Ideal Goethite |
|---|---|---|---|---|---|---|---|---|---|
| $Al_2O_3$ | 0.11 | 0.37 | 0.26 | 0.10 | 0.35 | 0.18 | 0.25 | 0.54 | 0.00 |
| $SiO_2$ | 3.77 | 3.53 | 3.66 | 1.39 | 3.64 | 3.63 | 3.27 | na | 0.00 |
| $Cr_2O_3$ | 0.02 | 0.03 | 0.12 | 0.00 | 0.04 | 0.09 | 0.05 | na | 0.00 |
| MnO | 0.00 | 0.00 | 0.00 | 0.00 | 0.00 | 0.00 | 0.00 | 0.09 | 0.00 |
| $Fe_2O_3$ | 80.68 | 81.75 | 80.54 | 78.86 | 82.04 | 79.92 | 80.63 | 85.30 | 87.98 |
| CoO | 0.00 | 0.00 | 0.00 | 0.00 | 0.00 | 0.00 | 0.00 | na | 0.00 |
| NiO | 0.10 | 0.02 | 0.08 | 0.11 | 0.04 | 0.09 | 0.07 | na | 0.00 |
| CuO | 0.00 | 0.00 | 0.06 | 0.06 | 0.08 | 0.33 | 0.10 | na | 0.00 |
| BaO | 0.14 | 0.08 | 0.14 | 0.02 | 0.07 | 0.10 | 0.09 | na | 0.00 |
| $H_2O$ | 15.19 | 14.21 | 15.14 | 19.47 | 13.74 | 15.66 | 15.53 | 14.07 | 12.02 |
| Σ | 100.00 | 100.00 | 100.00 | 100.00 | 100.00 | 100.00 | 100.00 | 100.00 | 100.00 |
| | Oxygen normalized to 3 | | | | | | | | |
| Al | 0.004 | 0.013 | 0.009 | 0.004 | 0.012 | 0.007 | 0.009 | 0.020 | 0.000 |
| Si | 0.114 | 0.106 | 0.111 | 0.045 | 0.109 | 0.111 | 0.100 | 0.000 | 0.000 |
| Cr | 0.000 | 0.001 | 0.003 | 0.000 | 0.001 | 0.002 | 0.001 | 0.000 | 0.000 |
| Mn | 0.000 | 0.000 | 0.000 | 0.000 | 0.000 | 0.000 | 0.000 | 0.002 | 0.000 |
| $Fe^{3+}$ | 1.841 | 1.844 | 1.837 | 1.933 | 1.840 | 1.836 | 1.853 | 1.979 | 2.000 |
| Co | 0.000 | 0.000 | 0.000 | 0.000 | 0.000 | 0.000 | 0.000 | 0.000 | 0.000 |
| Ni | 0.002 | 0.001 | 0.002 | 0.003 | 0.001 | 0.002 | 0.002 | 0.000 | 0.000 |
| Cu | 0.000 | 0.000 | 0.001 | 0.001 | 0.002 | 0.008 | 0.002 | 0.000 | 0.000 |
| Ba | 0.002 | 0.001 | 0.002 | 0.000 | 0.001 | 0.001 | 0.001 | 0.000 | 0.000 |
| Σcation | 1.963 | 1.965 | 1.965 | 1.986 | 1.965 | 1.967 | 1.968 | 2.001 | 2.000 |
| Σcharge | 6.000 | 6.000 | 6.000 | 6.000 | 6.000 | 6.000 | 6.000 | 6.000 | 6.000 |
| $H_2O$ molecule | 1.265 | 1.183 | 1.261 | 1.621 | 1.144 | 1.304 | 1.293 | 1.172 | 1.000 |

$H_2O$ by difference; na: not analyzed.

### 6.6. Origin of Sulfide Inclusions in Diamond

Sulfide inclusions in diamond are common [1,63,64] and may occur as discrete crystals [65]. Primary sulfide inclusions in diamonds have been shielded from the interaction with the outside environment. Therefore, such sulfide inclusions provide information on the primary compositions of mantle sulfide, the distribution and abundance of chalcophile elements in the mantle, and the formation environment of the host diamond [66,67].

Sulfide inclusions in diamonds are associated with either peridotitic or eclogitic assemblages [59,65]. On the base of the inclusion assemblage, the sulfide inclusion in the diamond LN50D04, which contains chromite and olivine, belongs to the peridotitic suite. In Siberian, peridotitic sulfides in diamond show significantly higher Ni and Cu contents than eclogitic sulfides [59]. The boundary between peridotitic and eclogitic sulfide inclusions is 8 wt. % Ni [65], or 12 wt. % Ni, or it is transitional [59]. Experimental studies on Ni/Fe exchange between olivine and monosulfide solid solution [68–70] indicate that, for peridotitic olivine with $Fo_{88}$–$Fo_{94}$ and 2500–3500 ppm Ni, the coexisting monosulfide solid solution will contain 30–55 mol % NiS (25–35 wt. % Ni). The sulfide inclusion in the diamond LN50D42 has a Ni content of ~30 wt. % and was probably in equilibrium with mantle olivine. If sulfide were the only inclusion in diamond, it may be difficult to determine which assemblage a sulfide inclusion might belong to based on Ni content. For example, the peridotitic sulfide in the diamond LN50D04 have Ni content from 6 to 34.6 wt. % (Figure 16b), whereas an eclogitic sulfide in an omphacite- and coesite-bearing diamond contains >11 wt. % Ni [59]. Deines and Harris [71]demonstrated that assignment of Ni-rich monosulfide to the peridotitic paragenesis is ambiguous if there is no further evidence from coexisting silicate inclusions.

Iron–nickel–copper sulfides are the most abundant primary sulfide inclusions in diamond [53,59,72]. Based on the stability of diamond (1500 K at 50 kbar [73]) and silicate assemblages in diamond, sulfide was trapped at around 1000–1200 °C. At these temperatures, monosulfide solid solution and sulfide melt coexist over a wide range of compositions [74,75]. Therefore, iron–nickel–copper sulfides in diamonds might be trapped as droplets of primary, immiscible sulfide melt, such as those from silicate megacrysts in basalt [76], or as crystals of monosulfide solid solution which then exsolved to different sulfide after subsolidus re-equilibration [1,2,27,63,64,77–80] (. Element partitioning between sulfide melt and monosulfide crystal at mantle conditions is controlled by composition, temperature, and pressure. The effect of melt composition and temperature on element partitioning between sulfide melts and monosulfide crystals suggest that Cu and Ni are slightly concentrated in residual melt during fractional crystallization at 1000–1100 °C [81]. Therefore, monosulfide solid solution crystallized from melts would contain low Ni and Cu, whereas the residual melts would be enriched in Ni and Cu. According to Bulanova et al. [59], peridotitic sulfide inclusions with Cu > 3 wt. % and Ni > 17 wt. % may represent entrapped melts. The bulk composition of each sulfide inclusion in diamond from the No. 50 diatreme likely has low Cu content (<3 wt. %), although a few analyses of the sulfide inclusions have higher Cu contents (Table 6). The Ni contents of the sulfide inclusions are widespread, from ~3 to >30 wt. % (Table 6). The Cu and Ni contents appear to suggest that the most sulfide inclusions were trapped as monosulfide crystals and some may be trapped as melts.

### 6.7. Source Rocks, Metasomatism, and Diamond Formation

The mineral inclusions examined in this study belong to harzburgitic and lherzolitic suites. Lherzolitic and harzburgitic inclusions can be recognized on the base of chemical compositions of garnet. For example, on a CaO versus $Cr_2O_3$ diagram for garnet (Figure 14), a solid line distinguishes lherzolitic garnet field (lower right side) from harzburgitic garnet field (upper left side) [46]. The compositions of most garnet inclusions from the No. 50 diatreme falls within the harzburgitic field, indicating that the main source rock of the diamonds is harzburgitic, while the lherzolitic assemblage is minor. The dominant harzburgitic source is consistent with the types of mineral inclusions recovered in diamond. The present study shows that there are abundant olivine, chromite, orthopyroxene and garnet inclusions in diamond, but rare clinopyroxene inclusions. The harzburgitic mantle likely formed by earlier extensive partial melting, whereas the lherzolitic mantle may have experienced a smaller degree of partial melting because of the high Mg # of the olivine.

Although they come from a harzburgitic diamond source, olivine inclusions are slightly variable in composition. For example, the olivine inclusions have a somewhat lower Mg # (peak at 93, Figure 9a) than similar harzburgitic inclusions worldwide (Mg # peak at 94, [1]), indicating that the harzburgitic mantle source was either chemically less depleted than in most other cratons or had been re-enriched in iron. Another feature of olivine inclusions is their higher Ni content. The NiO content of the olivine inclusions vary from 0.25 to 0.45 wt. % with a peak around 0.40 wt. % (Table 2; Figure 9b), higher than that for the olivine inclusions worldwide (0.36 wt. %, [24]). One olivine inclusion contains as high as 0.8 wt. % NiO (Table 2). In addition, some chromite inclusions in the diamond from the No. 50 kimberlite contain up to 0.79 wt. % $TiO_2$, higher than the maximum $TiO_2$ content (0.65 wt. %) of chromite inclusions in diamond worldwide. The above features, plus the existence of carbonate inclusions and the possible presence of hydrous silicate phases in diamond, imply a metasomatic enrichment event in the source region of diamond beneath the North China craton. We suggest that the diamond from the No. 50 kimberlite diatreme likely formed by growth under metasomatic conditions with the presence of a fluid.

**Funding:** This study was supported by funding from NSF (EAR 93-15918, 94-58368 and 97-25566 to YZ, EAR 91-17772 and 95-26596 to EJE), NWT Geology Division of DIAND of the Government of Canada, the University of Michigan, and the Geological Society of America. The electron microprobe analyzer used in this work was acquired under Grant # EAR-82-12764 from the National Science Foundation.

**Acknowledgments:** The diamond samples were provided by or purchased from the Sixth Geological Exploration Team of Liaoning Province. The field trip was sponsored by the Institute of Mineral Resources of Chinese Academy of Geological Sciences and the Liaoning Sixth Geological Team. Yunhui Huang, Shuying Qin, Zhuguo Han, Yawen Cao, Ruishan Liu, Jize Lin, Qing Miao, Liping Wang, Weixin Wang, and Xianfeng Fu are thanked for providing samples or for assistance for facilitating the field trip. This study was supervised by Eric J. Essene and Youxue Zhang in Department of Geological Sciences at the University of Michigan. Two anonymous reviewers are thanked for their helpful and constructive comments.

**Conflicts of Interest:** The authors declare no conflict of interest.

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
