# Peer review of "Electron Probe Microanalysis and Microscopy of Polishing-Exposed Solid-Phase Mineral Inclusions in Fuxian Kimberlite Diamonds"

_minerals, doi:10.3390/min12070844_

Round 1
Reviewer 1 Report
The present work focuses on the study of the inclusions present in diamonds from a Kimberlitic diatreme in China; through the study of their chemical composition and morphology, the author wants to investigate the origin of these inclusions and of the hosting diamonds. The work is certainly very interesting and worthy of publication but I think it could be improved with some small changes.
In the paragraph on Materials and methods, the Raman methodology used for the analyses should be described in greater detail.
In my opinion the final two paragraphs should be included in a final chapter of Conclusions. The Conclusions could also be broadened and they should include all the analyzes presented for the work. As a matter of fact In the current version the conclusions seem a little reduced.
I also believe that the English form could be improved: in this regard I have made some corrections that I report in the attached pdf file.

Reviewer 2 Report
This article is devoted to a detailed study of the chemical composition of mineral phases included in diamonds from kimberlite pipe No. 50 of Liaoning Province, China.
New data on inclusions in diamonds contribute to the understanding of the processes occurring in the mantle associated with the formation of diamonds. Some of them are common to all cratons, while others, on the contrary, reflect the local features of mantle blocks. Therefore, this article draws the attention of the reader by setting the problem itself.
The author has studied a large representative collection of rare material, carried out a thorough labor-intensive sample preparation, carried out analytical work at a good level, and the significance of the data obtained is beyond doubt. The data presented on this object by previous researchers harmoniously fit into the canvas of the narrative and, together with the data obtained by the author, form a general picture, on the basis of which the final conclusions are made. However, I would like to recommend the author to formulate the main conclusions more clearly (in a thesis form).
There are a number of points to which it is necessary to draw the attention of the author.
First, a presentation of research methods. In section 4. Analytical methods, the author lists a wide range of modern methods (backscattered electron (BSE) imaging from SEM and EPMA, EDS X-ray mapping, EPMA quantitative chemical analysis, Fourier-transform infrared spectros-copy (FTIR), micro-Raman spectroscopy ), however the description is given only for EPMA. In other sections, fragmentary descriptions of a particular method are given without indicating the names of the instruments used. I would like all the methods used to be described in detail in the appropriate section.
Secondly, thermobarometry. Apparently, it was originally intended, since the List of References contains a whole block of references to works on this subject. These references do not appear in the text. In this regard, it is necessary either to remove unnecessary references, or to calculate the P-T parameters where possible and load the conclusions with an additional genetic component. The same applies to unused references in the text to works on the harzburgite genesis of diamonds.
Thirdly, since we are talking about links. The two references mentioned in the text are not in the bibliography. In addition, in the corrected text, I took the liberty of recommending to the author to look at several articles of the last decade, which, in my opinion, could be very useful for argumentation and decorate the article.
In general, the article gives the impression of a good study and, of course, I recommend it for publication, taking into account the comments and making the necessary amendments.

Author Response
Please see the attachment submitted to Reviewer 1 for my response to the comments of the two reviewers.
Uploaded here is the Word format of the revised manuscript.
